# Skill-it! A data-driven skills framework for understanding and training language models

**Mayee F. Chen**[*]
Stanford University

**Nicholas Roberts**
University of Wisconsin-Madison

**Kush Bhatia**
Stanford University

**Jue Wang**
Together AI

**Ce Zhang**
Together AI, University of Chicago

**Frederic Sala**
University of Wisconsin-Madison

**Christopher Ré**
Stanford University

## Abstract

The quality of training data impacts the performance of pre-trained large language models (LMs). Given a fixed budget of tokens, we study how to best select data that leads to good downstream model performance across tasks. We develop a new framework based on a simple hypothesis: just as humans acquire interdependent skills in a deliberate order, language models also follow a natural order when learning a set of skills from their training data. If such an order exists, it can be utilized for improved understanding of LMs and for data-efficient training. Using this intuition, our framework formalizes the notion of a skill and of an ordered set of skills in terms of the associated data. First, using both synthetic and real data, we demonstrate that these ordered skill sets exist, and that their existence enables more advanced skills to be learned with less data when we train on their prerequisite skills. Second, using our proposed framework, we introduce an online data sampling algorithm, SKILL-IT, over mixtures of skills for both continual pre-training and fine-tuning regimes, where the objective is to efficiently learn multiple skills in the former and an individual skill in the latter. On the LEGO synthetic in the continual pre-training setting, SKILL-IT obtains 37.5 points higher accuracy than random sampling. On the Natural Instructions dataset in the fine-tuning setting, SKILL-IT reduces the validation loss on the target skill by 13.6% versus training on data associated with the target skill itself. We apply our skills framework on the RedPajama dataset to continually pre-train a 3B-parameter LM, achieving higher accuracy on the LM Evaluation Harness with 1B tokens than the baseline approach of sampling uniformly over data sources with 3B tokens.

## 1 Introduction

Large language models (LMs) exhibit remarkable capabilities, including producing creative content [55], writing source code [9], and chatting with users [8]. A key ingredient in enabling models to perform such tasks is the data on which the models are trained [18, 20, 59]. A natural way to unlock particular capabilities is to improve this training data. However, it is unclear how to select data from a large corpus for these capabilities given a fixed budget of training tokens, as data selection methods for current state-of-the-art LMs mostly rely on heuristics for filtering and mixing together different datasets [33, 59]. We lack a formal framework for capturing how data influences the model's capabilities and how to utilize this data effectively for improving LM performance.

---

[*]Correspondence to `mfchen@cs.stanford.edu`.

37th Conference on Neural Information Processing Systems (NeurIPS 2023).

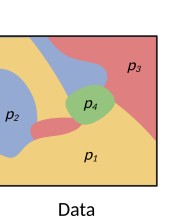 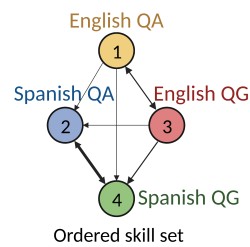 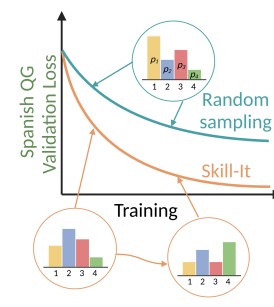

Data   Ordered skill set

Figure 1: Inspired by how humans acquire knowledge, we hypothesize that LMs best learn skills in a particular order and that this can help improve our understanding and training of LMs. We show that these ordered skill sets exist in real data, which enables skills to be learned with less data given that we train on their prerequisite skills. We then propose SKILL-IT, an online data selection algorithm that learns skills quickly by exploiting their ordering.

To develop such a framework, we take inspiration from how humans acquire knowledge. A classic idea in education literature is the concept of *skills* that form a learning hierarchy [65]. For example, one study found that students learned mathematical and scientific skills most quickly when these skills were presented in a particular order [12]. We seek to understand the extent that similar skill-based orderings characterize LM training. Such orderings, if they exist, may provide a better understanding of LMs as well as a mechanism for data-efficient training. For instance, to train an LM for Spanish question generation, we wish to know if training first on related but simpler tasks, such as Spanish grammar and English question generation, helps.

We study if the idea of skill orderings can help us build a framework that relates data to LM training and behavior. This requires addressing two challenges revolving around the connection between skills and data. First, in order to show that there exist sets of skills that the LM learns most efficiently in some particular order, *an operational definition of LM skill and skill ordering must be developed and validated on data*. In initial experiments, we investigated if semantic groupings of data, such as metadata attributes or embedding clusters, were sufficient to represent a skill and characterize how models learn. For instance, we partitioned the Alpaca dataset [56] by instruction type—a technique used to capture dataset diversity [62]—but we found that sampling based on instruction types and random sampling resulted in similar model performance, suggesting that not just any existing notion of data groups can characterize skills.

Second, *these definitions of skills must be used to construct sampling distributions to actually improve model training*. To develop criteria for a data selection algorithm that learns skills efficiently, we identify challenges that naive selection approaches face. The standard approach of random uniform sampling over data fails to learn skills optimally due to not accounting for skill imbalance and ordering. Skills can be distributed unevenly in the data, with more complex skills being rare—for instance, Spanish and question generation (QG) are $5\%$ and $4\%$ of the Natural Instructions dataset [63], respectively, but Spanish QG is only $0.2\%$. Random sampling also provides no mechanism for taking into account a particular training order and dependency structure on skills. More sophisticated techniques like curriculum learning account for sample-level ordering, but not skills or their dependencies. Our goal framework must account for these issues of imbalance and ordering.

**Skill-based framework** We define a *skill* as a unit of behavior that a model can learn using an associated slice of data (Definition 1). An *ordered skill set* is a collection of skills with a directed *skills graph* that is neither complete nor empty, where an edge from a prerequisite skill to a skill exists if the amount of training it takes to learn the skill can be reduced if the prerequisite skill is also learned (Definition 2, Figure 1 left, center). We show that ordered skill sets exist in synthetic and real datasets using this operational definition. Interestingly, the existence of these ordered skill sets unveils that one can learn a skill quickly not by training solely on that skill, but on a mixture of that skill and prerequisite skills. For instance, in Figure 3 we observe that Spanish QG can be learned more efficiently when the model also learns English QG and Spanish—we can achieve $4\%$ lower validation loss than training on only Spanish QG over a fixed budget of overall training steps.

Next, given an ordered skill set to train on, we use our framework to propose methods for how to select data so that the LM learn skills faster: skill-stratified sampling and an online generalization, SKILL-IT. We address the issue of unevenly distributed skills in datasets by proposing skill-stratified sampling, a

simple approach that allows us to explicitly optimize for learning skills by uniformly sampling relevant skills (such as a target skill and its prerequisite skills in fine-tuning). Skill-stratified sampling uses the construction of the ordered skill set but is static, which does not incorporate the ordering as training proceeds and results in oversampling skills that may be already learned early on in training. We address this issue by proposing an online data selection algorithm, SKILL-IT, for selecting mixtures of training skills that allocates more weight towards learning skills that are not yet learned or towards prerequisite influential skills (Figure 1 right). SKILL-IT is derived from an online optimization problem over the training skills for minimizing loss on a set of evaluation skills given a fixed budget of data and the skills graph. SKILL-IT is inspired by online mirror descent and can be adapted for continual pre-training, fine-tuning, or out-of-domain evaluation depending on the relationship between the evaluation skill set and the training skill set.

We evaluate SKILL-IT on synthetic and real datasets at two model scales, 125M and 1.3B parameters. For the continual pre-training setting, we show on the LEGO synthetic [72] that we obtain a 37.2 point improvement in accuracy over randomly selecting training data and a 9.7 point improvement over curriculum learning [4]. For the fine-tuning setting, we show that on the widely-used Natural Instructions dataset [41, 64], our algorithm over a mixture of skills is able to achieve up to 13.6% lower loss on that skill than solely training on that skill, given the same overall training budget. For the out-of-domain setting when our training skills do not align perfectly with evaluation skills, our algorithm is able to achieve the lowest loss on 11 out of 12 evaluation skills corresponding to task categories in the Natural Instructions test tasks dataset over random and skill-stratified sampling on the training data. We finally apply our framework to a case study on the recent RedPajama 1.2 trillion token dataset [57]. We use the data mixture produced by SKILL-IT to continually pre-train a 3B parameter model. We find that SKILL-IT achieves higher accuracy with 1B tokens than uniform sampling over data sources with 3B tokens.

## 2    Related work

An extended related work can be found in Appendix B. Existing work on data selection for LMs has generally ranged from more computationally expensive methods for dataset condensation on smaller datasets [47, 58, 48] to broader deduplication and filtering techniques for web-scale datasets [1, 33, 69]. Another way of improving model performance through choice of data is via curriculum learning [4], which also draws inspiration from how humans learn and arranges data in order from easiest to hardest over samples or groups [60]. In contrast to existing works in both curriculum learning and data selection, our work focuses on selecting data for learning an ordered set of skills more efficiently. How LMs learn is also a topic of growing interest; one framework posits that models learn over quanta, discrete units of computation [38], and another proposes that scaling laws for LMs can be understood in terms of learning combinations of skills [2]. Lastly, the notion of skill has been studied in education, ranging from classical research on learning hierarchies [66] to methods for decision-making over lesson sequences [49].

## 3    Skills framework

First, we propose definitions of skills and ordered skill sets in order to formalize our intuition around how models learn skills, and we demonstrate that not just any existing notion of data groups can characterize an ordered skill set in the dataset. Then, we demonstrate the existence of ordered skill sets on synthetic and real data, which show how viewing data through a skills-based framework can help with training and understanding model performance. Finally, we explore unsupervised skill recovery from data, finding that embedding-based approaches do not adequately recover synthetic skills.

### 3.1    Definitions

We first present a definition of an individual skill. Let the input space of all possible text data be $\mathcal{X}$, where $x \in \mathcal{X}$ is an individual text sample that a next-token-prediction LM $f \in \mathcal{F} : \mathcal{X} \to \mathcal{X}$ is trained on. We quantify learning via a metric $L : \mathcal{F} \times \mathcal{X} \to \mathbb{R}$, which maps from a model and evaluation data to a scalar quantity. In our setup, we use the cross-entropy validation loss applied over next-token predictions as our metric $L$.

**Definition 1 (Skill)** *A skill $s$ is a unit of behavior with associated data $\mathcal{X}_s \subseteq \mathcal{X}$ such that if $f$ is trained on an dataset $\mathcal{D}_s \subset \mathcal{X}_s$, $f$ has improved metric $L$ afterwards on samples belonging to $\mathcal{X}_s \setminus \mathcal{D}_s$ on average.*

This definition of a skill is flexible—it simply means that given a training dataset associated with the skill, a model $f$ has an improved metric (e.g., decreasing validation loss) when evaluated on validation

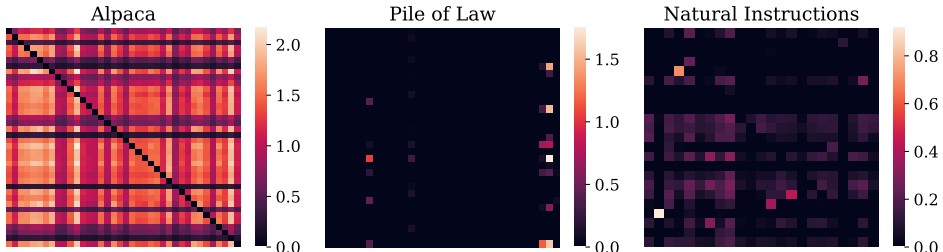

Figure 2: Heatmaps of adjacency matrices we compute for skill graphs for Alpaca, Pile of Law, and Natural Instructions. Negative elements and diagonals are thresholded to 0 for clarity. See Appendix D.2 for descriptions of how they were constructed and larger versions.

data associated with this skill. Under this definition, a skill could be a granular task, such as Spanish question generation for a subset of Wikipedia articles, or can be defined over a data source, such as next-token prediction of legal data from tax court rulings. However, our next definition, the ordered skill set, has a more specific construction and provides a framework for how models learn across dependent skills.

**Definition 2 (Ordered skill set, skills graph)** *An ordered skill set for $f$ is a collection of skills $\mathcal{S} = \{s_1,...,s_k\}$ over which there is a directed skills graph $G = (\mathcal{S},E)$ on the skill set that is neither complete or empty, where $(s_i,s_j) \in E$ if the amount of data needed to learn $s_j$ when uniformly sampling from $\mathcal{D}_{s_i} \cup \mathcal{D}_{s_j}$ is no more than the amount of data needed when sampling only from $\mathcal{D}_{s_j}$. We equate learning a skill $s_j$ to $f$ attaining a certain value of $L$ or lower on average over $\mathcal{X}_{s_j} \setminus \mathcal{D}_{s_j}$.*

This definition isolates complete and empty graphs as extrema that do not capture meaningful sets of skills. We discuss the three types of skill graphs—complete, empty, intermediate—and their implications for data selection. In particular, we discuss how several initial attempts of defining skills over datasets via semantic groupings resulted in the extrema cases (see Appendix D.2 for full results):

- The complete graph demonstrates that all skills influence each other. A random partition is an example of a skill set that yields a complete graph. This graph suggests that the best approach for learning any skill or set of skills is random sampling on the dataset. This is not a setting where we can gain much with skill-based sampling. For example, using instruction types as skills on the Alpaca dataset results in a nearly complete estimated skills graph (97.4% dense), and we find that stratified sampling on these skills only improves validation loss per skill by 0.007 points over random sampling on average (Figure 2 left), suggesting that utilizing skills does not improve model performance in this case.
- The empty graph demonstrates that each skill is independent. This can occur if skills are too granular; for instance, learning Spanish math problems is unlikely to help with English poem generation. This graph suggests that the best approach for learning an individual skill is to train on the skill itself. We see that empty graphs exist in real data; in Figure 2 (center), using data sources as skills on the Pile of Law [22] results in a nearly empty skills graph (3.9% dense).
- Graphs that are neither empty nor complete thus suggest a nontrivial order of how skill influence each other. *This is the setting in which we expect that identifying skills and exploiting their ordering will help the most.* In Figure 2 right, we use task categories, which capture broader reasoning patterns, as skills on Natural Instructions and find that the estimated graph has intermediate density (42.7% dense). We show concrete examples of how skills can be learned more efficiently on Natural Instructions in Section 3.2.

While these intuitive groupings result in ordered skill sets on some datasets (e.g., task categories on NI), this is not always the case (e.g., instruction types on Alpaca and sources on Pile of Law). Even though these groupings capture some notion of diversity in the dataset, our findings suggest that not just any semantic grouping induces an ordered skill set. We now empirically demonstrate that our definition of ordered skill sets aligns with how models learn and can be exploited for more data-efficient training.

## 3.2 Examples of skills and ordered skill sets

We provide examples of ordered skill sets on the LEGO synthetic dataset, an addition synthetic dataset, and subsets of the Natural Instructions dataset. On these datasets, we find that certain skills are better learned when trained along with their prerequisite skills rather than in isolation.

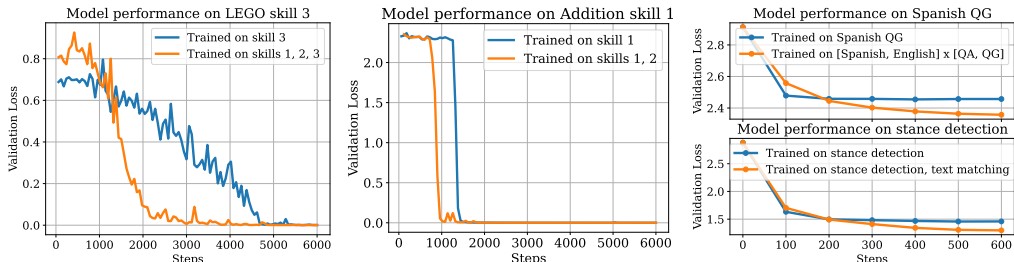

Figure 3: On the LEGO synthetic, 3-digit addition, and Natural Instructions, we identify examples of ordered skill sets in which training on a mixture of skills helps learn an individual skill faster than just training on that skill itself, given a fixed training budget.

**LEGO skills** The LEGO synthetic, first introduced in [72], can evaluate a model's ability to follow a chain of reasoning. In this synthetic, the letters of the alphabet, $\mathcal{A}$, are variables each with some binary label in $\{0,1\}$. An individual sample consists of $k$ clauses for some fixed $k$ across the dataset, each of the form $a = gx$ where $a, x \in \mathcal{A}$ and $g$ is either a negation ("not") or assertion ("val"), e.g. we assign $a$ to the value of $x$, or we assign $a$ to the opposite label. At the end of the sentence, we prompt the model for what the value of one of these variables is. Two samples $x \in \mathcal{X}$ are given below for $k = 5$:

> Input: b = not y, r = val 1, m = val b, q = val m, y = not r. Output: b = 1.
> Input: c = val x, p = val f, x = val k, f = not c, k = val 0. Output: k = 0.

These samples each correspond to a chain of reasoning; for instance the first sample has the chain $r, y, b, m, q$, where knowing $q$'s label requires the most reasoning steps. We define the $i$th skill $s_i$ as the model's ability to know the $i$th variable of the chain. From our example above, the first sample belongs to $\mathcal{X}_{s_3}$ and the second sample belongs to $\mathcal{X}_{s_1}$. To demonstrate the existence of ordered skill sets, we continually pre-train the 125M parameter GPT-Neo model [14, 6] over various mixtures of LEGO skills with $k = 5$. In Figure 3 (left), we find that in 35.9% fewer training steps, training on a balanced mixture of $\mathcal{X}_{s_1}, \mathcal{X}_{s_2}$, and $\mathcal{X}_{s_3}$ resulted in the same validation loss of $0.01$ as training solely on $\mathcal{X}_{s_3}$. This suggests that $s_1, s_2$ helped unlock performance on $s_3$ and that there exist edges from $s_1$ or $s_2$ to $s_3$ in the skill graph. Additional observations are available in Appendix E.1, where we examine other edges as well as more complex reasoning chains, and the full skills graph corresponding to the ordered skill set for LEGO with $k = 5$ is in Figure 11.

**Addition skills** We consider a variant of a synthetic 5-digit addition dataset analyzed in [45]. We show the existence of ordered skill sets for a simplified 3-digit addition dataset where we treat each digit prediction as a skill—the outputs, in this case, are the integers $\{0, 1, ..., 9\}$. Examples are of the following form:

> Input: A = 1 0 6 + 0 7 1, A 0 = ? Output: 7     Input: A = 6 0 6 + 8 7 9, A 2 = ? Output: 4

where 'A 0' refers to the ones digit of the output ($s_1$) and 'A 2' refers to the hundreds digit ($s_3$). In Figure 3 (center), we find that in 32% fewer training steps, training on a balanced mixture of $\mathcal{X}_{s_1}$, and $\mathcal{X}_{s_2}$ resulted in the same validation loss of $0.01$ as training solely on $\mathcal{X}_{s_1}$. That is, the ones digit addition skill can be improved by simultaneously learning the tens digit addition skill, even though the former should not require information from the latter—this is in line with observations from prior work that models do not always learn the ones digit addition first [45]. The full skills graph corresponding to the ordered skill set over 3-digit addition is in Figure 12.

**Natural Instructions (NI) skills** We show that ordered skill sets exist in NI [63] when we treat task categories as skills.

- In Figure 3 (top right), we show that ordered skill sets exist over crosslingual task categories. Training on Spanish question generation (QG) along with equal parts of English QG, Spanish question answering (QA), and English QA results in 4.1% lower validation loss than training only on Spanish QG. Remarkably, the former only uses 25% of the latter's Spanish QG data. This suggests that there are edges from Spanish QA, English QA, and English QG to Spanish QG.

- In Figure 3 (bottom right), we see that training on the task category Text Matching along with Stance Detection helps decrease the loss on Stance Detection by 11%. This suggests that these categories, which both involve understanding the relationship between two input texts, share an edge.

Table 1: Summary of three settings—continual pre-training, fine-tuning, and out-of-domain. These settings are determined by how $\mathcal{S}_{\text{eval}}$ is defined and result in different skills graphs used for our sampling methods.

| Setting | $\mathcal{S}_{\text{eval}}$ | Skills graph |
|---|---|---|
| Continual pre-training | $\mathcal{S}_{\text{eval}} = \mathcal{S}_{\text{train}}$ | $A \in \mathbb{R}^{k \times k}$, edges among all $\mathcal{S}_{\text{train}}$ |
| Fine-tuning | $\mathcal{S}_{\text{eval}} \subset \mathcal{S}_{\text{train}}$ | $A \in \mathbb{R}^{k \times m}$, edges from training skills to target subset |
| Out-of-domain | $\mathcal{S}_{\text{eval}} \cap \mathcal{S}_{\text{train}} = \emptyset$ | $A \in \mathbb{R}^{k \times m}$, edges from training skills to evaluation skill set |

The full skills graphs corresponding to the ordered skill sets over these task categories are in Figure 14. While equating task categories to skills may be noisy, these examples suggest that there is signal within real data that suggests that ordered skill sets can improve data efficiency.

### 3.3 Skill recovery

A final component of characterizing skills is unsupervised recovery of ordered skill sets. We consider embedding-based clustering approaches and a loss-based clustering approach for recovering LEGO skills. When clustering data using various trained and pre-trained embeddings, we find that they were unable to achieve above $39\%$ accuracy on LEGO. Instead, we find that taking 10 random training runs and clustering data by their *loss* per timestep per run recovers the skills with $61\%$ accuracy (Table 3). The intuition behind this method is that the validation losses on points from the same skill have similar trajectories as models learn. We discuss this approach more in Appendix E.2.

## 4 Skills-based data selection

Now that we have established the existence of ordered skill sets, we discuss how to use them for data selection. We state the data selection problem for learning across skills in Section 4.1. We discuss how to learn the skills graph that will be exploited in our data selection methods in Section 4.2. We then introduce two sampling methods that utilize the graph, a simple skill-stratified sampling method and the online sampling method SKILL-IT, in Section 4.3.

### 4.1 Problem statement

We are given an ordered training skill set $\mathcal{S}_{\text{train}} = \{s_{\text{train},1}, ..., s_{\text{train},k}\}$ on the training data, each with associated support set $\mathcal{X}_{s_{\text{train},1}}, ... \mathcal{X}_{s_{\text{train},k}}$, and an ordered evaluation skill set $\mathcal{S}_{\text{eval}} = \{s_{\text{eval},1}, ..., s_{\text{eval},m}\}$ of $m$ evaluation skills on a separate evaluation dataset. We aim to select $n$ samples from $\mathcal{S}_{\text{train}}$ via a mixture of training skills, $p \in \Delta^{k-1}$, to achieve three goals depending on how $\mathcal{S}_{\text{eval}}$ is constructed:

- **Continual pre-training**: when $\mathcal{S}_{\text{eval}} = \mathcal{S}_{\text{train}}$, our goal is select a mixture of training skills to learn all of them.
- **Fine-tuning**: when $\mathcal{S}_{\text{eval}} \subset \mathcal{S}_{\text{train}}$, our goal is to select a mixture of training skills to learn an individual target skill or subset of these skills.
- **Out-of-domain**: when $\mathcal{S}_{\text{eval}} \cap \mathcal{S}_{\text{train}} = \emptyset$, our goal is to select a mixture of training skills to learn a disjoint set of evaluation skills we cannot train on. This can arise when we have a separate downstream validation dataset or the skills identified in the training dataset are noisy.

Furthermore, we have a skills graph $G = (\mathcal{S}_{\text{train}} \cup \mathcal{S}_{\text{eval}}, E)$, where $E \subseteq \mathcal{S}_{\text{train}} \times \mathcal{S}_{\text{eval}}$ and $A \in \mathbb{R}^{k \times m}$ is a weighted adjacency submatrix, where $A_{ij}$ describes the strength of the edge from $s_{\text{train},i}$ to $s_{\text{eval},j}$. In Table 1, we summarize how the three different settings are constructed and how $A$ varies across them. Next, we discuss how $A$ can be estimated from the data.

### 4.2 Skills graph learning

The skills graph is important for determining how to sample from the ordered skill set for training efficiently. We present two approaches for learning the skills graph—brute-force and linear approximation. Algorithms are provided in Appendix C.2. By definition 2, the brute-force way of identifying edges involves fixing an overall training budget of $H$ steps and 1) training and evaluating the model on each $s_{\text{train},i}$ and 2) training the model on each pair of $(s_{\text{train},i}, s_{\text{eval},j})$ and evaluating on $s_{\text{train},i}$ and $s_{\text{eval},j}$. If the loss on $s_{\text{eval},j}$ when trained on both $s_{\text{train},i}$ and $s_{\text{eval},j}$ is lower, there exists an edge from $s_{\text{train},i}$ to $s_{\text{eval},j}$ with edge weight proportional to the difference in loss. This approach has runtime $\mathcal{O}(Hkm)$ and is only feasible when $k$ is small and when we have access to $\mathcal{S}_{\text{eval}}$ at training

---

**Algorithm 1** SKILL-IT Online Data Selection Algorithm

---

1: **Input:** Ordered training skill set $\mathcal{S}_{\text{train}}$, ordered evaluation skill set $\mathcal{S}_{\text{eval}}$. Learning rate $\eta$, $T$ rounds, $n$ samples, $H$ training steps per run for graph learning, model $f_1$, window parameter $w$.
2: $A \leftarrow \text{LEARNGRAPH}(\mathcal{S}_{\text{train}}, \mathcal{S}_{\text{eval}}, H, f_1)$ (Alg. 2, 3).
3: Initialize $p_1^i = \exp(\eta \sum_{j=1}^m A_{ij})$ for all $i \in [k]$, the softmax over $A$.
4: **for** $t = 1, ..., T-1$ **do**
5:      Observe losses $L_{\text{eval},j}(f_t)$ for all $s_{\text{eval},j} \in \mathcal{S}_{\text{eval}}$.
6:      Train model $f_t$ with $n/T$ samples from mixture $p_t$ over $\mathcal{S}_{\text{train}}$. Update model $f_{t+1} = \Phi(f_t, p_t)$.
7:      Set $p_{t+1}^i = \exp(\eta \sum_{\tau=t-w+1}^t \sum_{j=1}^m A_{ij} L_{\text{eval},j}(f_\tau))$.
8: **end for**

---

time. Otherwise, we can approximate this approach in linear time by training on each $s_i$ for $h < H$ steps and setting $A_{ij} > 0$ if the loss on $s_j$ decreases over $h$ steps for a runtime of $\mathcal{O}(hk)$. This linear approach is necessary in the out-of-domain setting, since it does not require training on $\mathcal{S}_{\text{eval}}$. In addition, both graph learning approaches can be performed on a smaller model, and the learned graph can be used for data selection for training a larger model (Appendix E.4).

### 4.3 Skills graph-aware sampling

We present two approaches for sampling over the mixture of training skills according to the skills graph: skill-stratified sampling, which samples uniformly over relevant training skills according to $A$, and SKILL-IT, an online generalization that incorporates feedback of how skills are being learned so far.

#### 4.3.1 Skill-stratified sampling

A straightforward sampling approach is to discard training skills that do not benefit the evaluation skills and sample uniformly over the set of relevant training skills, which we call *skill-stratified sampling*. For continual pre-training, the relevant skills are the entire training skill set; for each $s_{\text{train},i} \in \mathcal{S}_{\text{train}}$, $\Pr(s_{\text{train},i}) = \frac{1}{k}$. This enables each skill to have sufficient training data. For fine-tuning, the relevant skills are the target skills and prerequisite skills, which can be identified via positive entries of the $i$th column of $A$ with $\mathcal{S}_{\text{prereq}} = \{s_{\text{train},i} : \exists\, s_{\text{eval},j} \text{ s.t. } A_{ij} > 0\}$. We then set $\Pr(s) = \frac{1}{|\mathcal{S}_{\text{prereq}} \cup \mathcal{S}_{\text{eval}}|}$ for $s \in \mathcal{S}_{\text{prereq}} \cup \mathcal{S}_{\text{eval}}$. For the out-of-domain setting, skill-stratified sampling is over the set of prerequisite skills. For each $s \in \mathcal{S}_{\text{prereq}}$, we set $\Pr(s) = \frac{1}{|\mathcal{S}_{\text{prereq}}|}$. Next, we propose our online algorithm that exploits the graph dynamically for more efficient training.

#### 4.3.2 SKILL-IT online data selection algorithm

Despite accounting for prerequisite skills, one shortcoming of skill-stratified sampling is that even if a skill has already obtained sufficiently low validation loss early during training, we will continue to allocate the same weight to that skill throughout training. Therefore, we formulate our data selection problem as an online learning problem and propose SKILL-IT, which both prioritizes prerequisite skills and skills that are not yet learned.

We are given a budget of $T$ rounds and $n$ total samples to train on. At round $t$, we select a mixture $p_t \in \Delta^{k-1}$ from the $k$-dimensional unit simplex, and for each training skill $s_{\text{train},i} \in \mathcal{S}_{\text{train}}$, we sample from $\mathcal{X}_{s_{\text{train},i}}$ with proportion $p_t^i$ for a total of $\frac{n}{T}$ samples per round. Let $f_t$ be the model at at the start of round $t$. We can define $f_t$ recursively as a function of the previous round's model $f_{t-1}$ and mixture $p_{t-1}$ via a dynamics function $\Phi : \mathcal{F} \times \Delta^{k-1} \to \mathcal{F}$; that is, $f_t = \Phi(f_{t-1}, p_{t-1})$. Let $L_{\text{eval},j}(f_t)$ be the validation loss of $f_t$ on $s_{\text{eval},j}$. Our goal is to select $p_1, ..., p_T$ to minimize loss per evaluation skill at the end of training:

$$\underset{p_1,...,p_T \in \Delta^{k-1}}{\text{minimize}} \frac{1}{m} \sum_{j=1}^m L_{\text{eval},j}(f_T). \tag{1}$$

This optimization problem is challenging to solve without additional assumptions. In order to make the problem tractable, we impose an explicit dynamics rule for the each evaluation skill's loss $L_{\text{eval},j}$ in terms of the current loss and data mixture. Assuming for simplicity that $\mathcal{S}_{\text{eval}} \subseteq \mathcal{S}_{\text{train}}$, a simple rule would be $L_{\text{eval},j}(f_t) = L_{\text{eval},j}(\Phi(f_{t-1}, p_{t-1})) := L_{\text{eval},j}(f_{t-1})(1 - \alpha p_{t-1}^j)$ for $\alpha \in [0,1]$. That is, we expect that allocating more data to skill $j$ should result in the validation loss on skill $j$ decreasing. However, such

an expression assumes that only training on the $j$th skill will help learn the $j$th skill. Instead, Section 3.2 suggests that there are other skills that may help with the $j$th skill. We propose the following dynamics:

$$L_{\text{eval},j}(f_t) = L_{\text{eval},j}(f_{t-1})(1 - A_{:,j}^\top p_{t-1}), \qquad (2)$$

where $A_{:,j}$ is the column with weights of all skills that influence $s_{\text{eval},j}$, and we absorb the scalar $\alpha$ into $A$. The optimization problem in (1) can thus be simplified as follows:

$$\underset{p_1,...,p_T \in \Delta^{k-1}}{\text{minimize}} \frac{1}{m}\sum_{j=1}^m L_{\text{eval},j}(f_T) \qquad (3)$$

$$\text{s.t } f_t = \Phi(f_{t-1}, p_{t-1}) \, \forall t = 1,...T$$

$$L_{\text{eval},j}(f_t) = L_{\text{eval},j}(f_{t-1})(1 - A_{:,j}^\top p_{t-1}) \, \forall j \in [m]$$

In Appendix C, we derive the following update rule via online mirror descent [46] for learning rate $\eta > 0$:

$$p_{t+1}^i = p_t^i \exp\left(\eta \sum_{j=1}^m A_{ij} L_{\text{eval},j}(f_t)\right). \qquad (4)$$

In addition, when equation 4 is expanded, we have that $p_{t+1}^i = p_1^i \exp\left(\eta \sum_{\tau=1}^t \sum_{j=1}^m A_{ij} L_{\text{eval},j}(f_\tau)\right)$. Since this summation over $\tau$ results in diminishing strength of updates, we change it to a moving window of size $w$. Our full method is in Algorithm 1.

Intuitively, at each step we adjust the weight on skill $i$ based on the losses of skills that $i$ influences, with the assumption that more training data helps decrease loss. Note that when we use our algorithm with a complete graph or empty graph, we achieve expected behavior discussed in Section 3.1. For the complete graph, our algorithm reduces to stratified sampling. When we have a skill set with an empty graph, the update rule reduces to sampling proportional to each skill's validation loss.

## 5 Experimental results

Given an ordered skill set, we aim to validate SKILL-IT's ability to select data for efficiently learning skills in the continual pre-training, fine-tuning, and out-of-domain settings. We provide full tables of results in Appendix E.3.1 and results where we learn the skills graph on the 125M model and use it for the 1.3B parameter model in Appendix E.4. Skills graphs are in Appendix D.2, weight trajectories for SKILL-IT are in Appendix E.3.2, and ablations on the graph and online components of SKILL-IT are in Appendix E.5.

### 5.1 Continual pre-training

**Setup** We evaluate the ability of SKILL-IT to select data for efficiently learning over all skills. We measure average validation loss per skill after a fixed number of training steps. We construct the LEGO synthetic and addition synthetic with $k = 5$ and $3$, respectively, and an imbalanced dataset over the skills. On the Natural Instructions dataset, we use 23 of the task categories as skills.

**Baselines** We compare SKILL-IT against three baselines that do not account for skills: random sampling, curriculum learning, and anticurriculum learning. Random sampling is a standard procedure for selecting samples given no additional information. Curriculum learning [4] and anticurriculum learning [67] score the samples from easiest to hardest and vice versa, respectively, and sample over an expanding set of the lowest scored samples at every epoch; we use the pre-trained model's loss to rank points. We evaluate skill-stratified sampling, which uses knowledge of the skills but is not online, and include an additional skills curriculum and anticurriculum baseline from [60], which samples from skills in order of average loss per skill.

**Analysis** Across our experiments we find that SKILL-IT outperforms baselines that do not use skills as well as skill-stratified sampling and skill curriculum learning. Our results on the LEGO dataset are shown in Figure 4. SKILL-IT and skill-stratified sampling attain lower loss than other approaches on skills 2, 3, 4 and on average, and while curriculum and anticurriculum learning attain lower loss on skill 5, they fail to learn other skills, resulting in at most 68 points accuracy on skill 3. SKILL-IT reaches

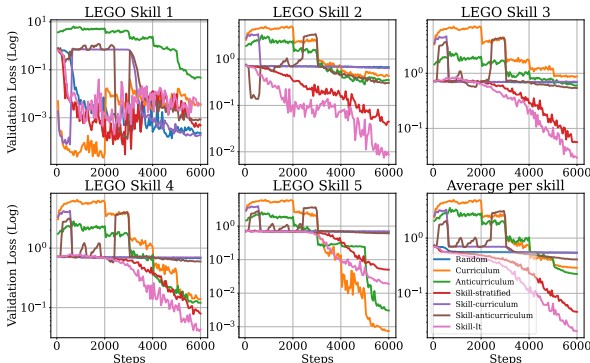

Figure 4: Performance of SKILL-IT on each skill in the continual pre-training setting (learning over all skills in the ordered training skill set) on the LEGO synthetic .

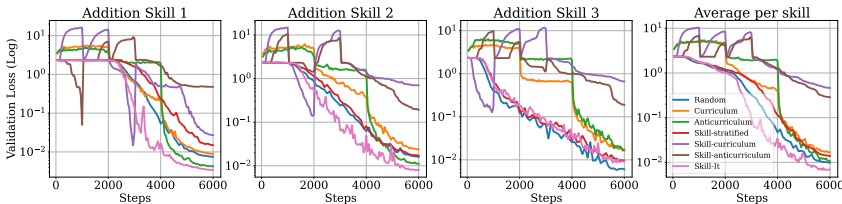

Figure 5: Performance of SKILL-IT in the continual pre-training setting on the addition synthetic.

a high average accuracy earlier in training than other approaches; halfway through training, SKILL-IT obtains between $8.3$ and $33.5$ points higher average accuracy than other approaches, reaching a final accuracy of $99.3$ (Figure 20). SKILL-IT initially allocates more weight to prerequisite skills such as skill 2 as suggested by Figure 11 and later on allocates more weights to the skills that are learned more slowly, such as skills 4 and 5 (Figure 22). On the addition synthetic with $k = 3$, SKILL-IT obtains lower validation loss than the baselines on skills 1 and 2 in Figure 5. While most approaches aside from skill curriculum learning eventually obtain $100\%$ accuracy on all skills, SKILL-IT requires less training to reach sufficiently high accuracy; halfway through training, SKILL-IT has accuracy between $8.7$ and $73.1$ points higher than other approaches (Figure 21). Finally on Natural Instructions, the average validation loss from SKILL-IT is $3.2\%$ lower than from random sampling (Table 7). Our results suggest that exploiting the construction and ordering of skills is critical to learning skills quickly.

## 5.2 Fine-tuning

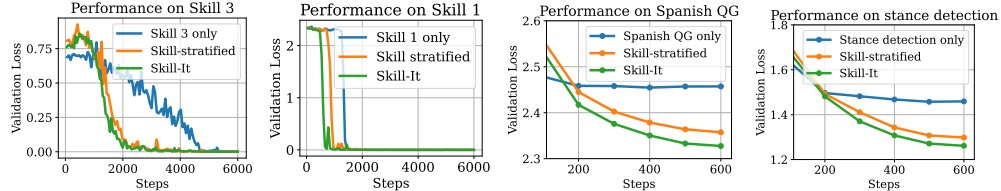

Figure 6: Performance of SKILL-IT in the fine-tuning setting on LEGO, addition, and NI.

**Setup** We evaluate the ability of SKILL-IT to select data from an ordered training skill set for learning a target skill. Mirroring Figure 3, we evaluate on LEGO target skill 3 (third in reasoning chain), on the addition synthetic's skill 1 (ones place digit addition), and on NI's Spanish QG and Stance Detection.

**Baselines** We compare SKILL-IT against training on the target skill only and skill-stratified sampling over prerequisite skills and the target skill. The skill-stratified sampling approach uses the ordered skill set to identify prerequisite skills, but does not exploit them dynamically.

**Analysis** Our results are shown in Figure 6. On LEGO, SKILL-IT results in the same validation loss of $0.01$ as training only on the target skill in $38.1\%$ fewer steps. We observe a similar trend on addition, with SKILL-IT converging to a validation loss of $0.01$ in $59\%$ fewer steps required to do so when training only on the target skill. Finally, on NI, SKILL-IT improves validation loss on Spanish question generation by $5.3\%$ and Stance Detection by $13.6\%$ over just training on the respective target skill only. In this setting,

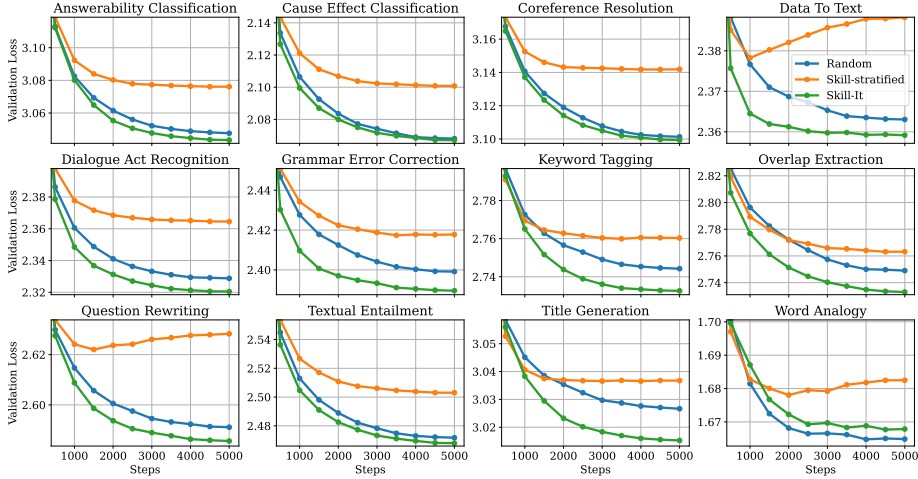

Figure 7: Performance of SKILL-IT in the out-of-domain setting for the NI test task split. SKILL-IT uses the graph between the train and evaluation skills to produce an online mixture on the training dataset.

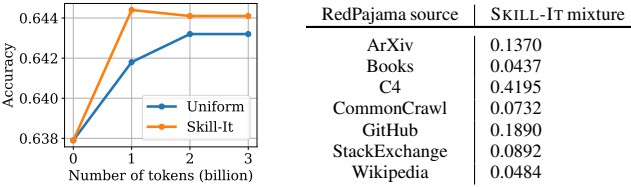

| RedPajama source | SKILL-IT mixture |
|---|---|
| ArXiv | 0.1370 |
| Books | 0.0437 |
| C4 | 0.4195 |
| CommonCrawl | 0.0732 |
| GitHub | 0.1890 |
| StackExchange | 0.0892 |
| Wikipedia | 0.0484 |

Figure 8: Left: Accuracy on LM Evaluation Harness for continual pre-training of a 3B parameter model using SKILL-IT on the RedPajama dataset. We achieve higher accuracy at 1B additional tokens than uniform at 3B tokens. Right: SKILL-IT mixture over RedPajama sources.

a significant portion of the improvement over training only on the target skill comes from identification of prerequisite skills through the learned graph in the skill-stratified sampling method. SKILL-IT is further able to improve performance with finer-grained dynamic weighting on prerequisite skills.

### 5.3 Out-of-domain setting

**Natural Instructions** We evaluate the ability of SKILL-IT to select data from a set of training skills for learning a disjoint set of evaluation skills that we cannot train on. We use all 59 task categories in the NI train tasks split as the training skills and the 12 task categories in the test tasks split as our evaluation skills. We compare SKILL-IT against random and skill-stratified sampling, both of which do not exploit the relationships between training skills and evaluation skills. SKILL-IT achieves the lowest loss on 11 out of 12 task categories over random and skill-stratified sampling (Figure 7, tables in Appendix).

**RedPajama** We use SKILL-IT to produce a data mixture on the RedPajama dataset. The training skills are the data sources comprising the dataset, and the evaluation skills are several tasks from the Language Model Evaluation Harness [15]. SKILL-IT with $T = 1$ yields the mixture in Figure 8 (right). We continually pre-train a 3B parameter model trained on 1T tokens for 3B additional tokens using this mixture, and see that it outperforms uniform sampling over the data sources (Figure 8 left). In particular, SKILL-IT achieves higher accuracy with 1B additional tokens than uniform with 3B additional tokens.

## 6  Conclusion

Given a fixed budget of data, knowing what data to train on to induce various capabilities in an LM is challenging. As LMs continue to improve, it will become increasingly important to extract as much signal as possible from the data and to direct that signal towards acquiring a broad variety of capabilities. In this paper, we introduce a skills-based framework for understanding how LMs learn and for selecting training data. We hope our study invites others to build on such a notion of skill and further explore how to align skills with data.

## Acknowledgements

We thank Together AI (https://together.ai/) and Stanford HAI for providing portions of the compute used to train models in this paper. We thank Sabri Eyuboglu, Karan Goel, Arjun Desai, Neel Guha, Michael Zhang, Vishnu Sarrukai, Simran Arora, Ben Spector, Brandon Yang, Gautam Machiraju, and Sang Michael Xie for their helpful feedback and discussion.

We gratefully acknowledge the support of NIH under No. U54EB020405 (Mobilize), NSF under Nos. CCF1763315 (Beyond Sparsity), CCF1563078 (Volume to Velocity), and 1937301 (RTML); US DEVCOM ARL under No. W911NF-21-2-0251 (Interactive Human-AI Teaming); ONR under No. N000141712266 (Unifying Weak Supervision); ONR N00014-20-1-2480: Understanding and Applying Non-Euclidean Geometry in Machine Learning; N000142012275 (NEPTUNE); NXP, Xilinx, LETI-CEA, Intel, IBM, Microsoft, NEC, Toshiba, TSMC, ARM, Hitachi, BASF, Accenture, Ericsson, Qualcomm, Analog Devices, Google Cloud, Salesforce, Total, the HAI-GCP Cloud Credits for Research program, the Stanford Data Science Initiative (SDSI), and members of the Stanford DAWN project: Facebook, Google, and VMWare. FS is supported by NSF CCF2106707 and the Wisconsin Alumni Research Foundation (WARF). The U.S. Government is authorized to reproduce and distribute reprints for Governmental purposes notwithstanding any copyright notation thereon. Any opinions, findings, and conclusions or recommendations expressed in this material are those of the authors and do not necessarily reflect the views, policies, or endorsements, either expressed or implied, of NIH, ONR, or the U.S. Government.

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

# A Broader Impacts and Limitations

**Broader Impacts**    As more LMs are developed, a key criteria for their adoption and utility is if they exhibit a wide array of useful capabilities, such as generating harmless content, summarizing essays, and being conversational with the user. While improvements in other parts of the LM development pipeline such as training and architecture are important, many recent advances in building LMs with a wide array of useful capabilities have come from the data itself [18, 56, 10, 16, 43]. Our work is fundamental in investigating how LMs learn and how to select data to learn skills more efficiently. However, we recognize that data selection methods can always be utilized to optimize for particular skills that may be considered malicious or negatively target or exclude specific groups [3]. Furthermore, pre-trained LMs have been found to have various biases [26, 44, 36, 7].

**Limitations**    The skills graph can either be provided (e.g., using a knowledge graph) or learned. Our work learns the skills graph using Algorithm 2 or Algorithm 3, which requires initial training runs on pairs of skills or each skill, respectively. This can be made more efficient by performing these training runs on a smaller model and for fewer number of steps, but tradeoffs here have yet to be thoroughly investigated. SKILL-IT also assumes that the ordered skill set is provided; as discussed in sections 3.1 and 3.3, it is challenging to recover ordered skill sets simply via metadata attributes or embedding clustering. Otherwise, the best way to sample over collections of skills that form a complete or empty graph is random or stratified sampling with no ordering to exploit. Our loss-based clustering approach presented in section 3.3 demonstrates that grouping by losses can provide an explanation for how skills are defined over data. An important direction for future work is to use such a clustering approach or other unsupervised algorithms in an end-to-end pipeline for skill discovery, skill graph learning, and data selection based on such skills.

**Code release**    The code for SKILL-IT is available at `https://github.com/HazyResearch/skill-it`.

# B Extended related work

**Data selection for LMs**    There have been several studies of large-scale data selection for LMs. Data deduplication [33, 1, 23], in which identical or nearly identical samples are removed, is a method that enables LMs to be trained on smaller, cleaned datasets and has been increasingly used as a pre-processing step for training data [59, 5, 71]. Other methods applied at scale involve ensuring high quality of data by explicitly filtering out samples or comparing the training dataset with a cleaned reference dataset [8, 59, 32]. Importance reweighting approaches have also been proposed for identifying training data from a large corpus that best approximates a smaller target distribution [69], and influence functions have been used to select a subset of training data to improve performance on downstream tasks [61]. These approaches can identify data pertaining to a particular target distribution or filter out low quality data according to some heuristic, while our work aims to understand how the choice of data is related to the numerous skills that LMs learn.

Recent development of LMs has shifted focus from emphasizing the scale of the model to prioritizing the training data utilized. For example, models like Alpaca [56], Vicuna [10], and Koala [16] are all based on the LLaMA model combined with instruction data generated by an existing LM. Palm 2's technical report states that the data mixture was a critical component of the final model [18], and Mosaic ML's recent MPT model was trained on a hand-engineered mixture of the RedPajama dataset [43]. However, these works lack rigorous explanation for why their training datasets were constructed in this way.

Finally, perhaps most related to our approach is the contemporary work DoReMi [68], which uses group distributionally robust optimization on a smaller LM to select data source mixtures for training a larger LM. Their approach focuses on selecting data at the data source level for optimizing worst-case performance across the training data sources, rather than at the more general skills level for a variety of target skill sets. Furthermore, we focus on understanding how skills are related to each other and induce some order in how LMs learn by explicitly modeling skill graph structure, which we find to be important for data-efficient LM training (see ablations in Appendix E.5).

**Data selection methods**    Many data selection methods have been proposed for supervised, task-specific settings. In this setting, the most typical objective is dataset condensation, which aims to identify a small subset of data that captures the larger dataset's properties with respect to the model.

Some approaches include constructing coresets [31, 48], identifying samples that the model forgets during training [58]; identifying samples with the largest gradients [47] or gradients that approximate the overall gradient [40]; clustering in embedding space and selecting points farthest from cluster centers [53]; and selecting samples with the highest uncertainty or entropy [34]. These approaches have also been shown to transfer from smaller models to larger models [11]. Unlike these methods, we study how to select data for learning one or many skills at the mixture level for LMs instead of the instance level.

Another area of interest is data selection for domain adaptation and multitask learning. For domain adaptation, there are a wide range of methods that select data to best match the target distribution. For example, the Moore-Lewis method matches data based on the difference in cross-entropy using a model trained on the target versus a model trained on the source data [42]. Several other approaches suggest training a model to distinguish between source and target and selecting points with high uncertainty [50], or selecting points based on some divergence in an embedding space [51]. In comparison to these approaches, our work focuses on learning one or many skills and also finds that embedding-based heuristics do not fully identify skills.

**Data attribution**    Another perspective on understanding training data is data attribution, which seeks to identify what data is responsible for particular model behaviors. Influence functions [29] and shapley values [17] are two ways to quantify the role of individual samples. Datamodels [24] fit a model to predict behavior given a subset of training data, providing a framework for understanding individual samples as well as dataset counterfactuals. Simfluence [21] fits a Markov process to a set of training trajectories for finer-grained understanding of how data impacts training. We focus on understanding how groups of data associated with skills elicit broader model capabilities, and utilize this understanding to select data for more efficient training.

**Curriculum learning**    Curriculum learning [4] proposes to show the model data in order from easy samples to hard ones. Various criteria have been used to determine hardness, and anticurriculum as well as various pacing functions and mixing rates have been explored [54]. Curriculum learning can also be performed at the group level [60]. More sophisticated approaches include parametrizing each sample with a dynamic importance [52], and also accounting for irrelevant and noisy data [39]. Our approach similarly utilizes a curriculum, but it is defined over a skills graph and does not necessarily align with training on easiest to hardest skills.

**How LMs learn**    Many different explanations for how LMs learn from data have been proposed. One hypothesis is that there exist discrete, universal building blocks of LM knowledge called quanta, and power law scaling emerges from a learning over a particular distribution of quanta in the right order [38]. Another work proposes that scaling laws for LMs can be understood in terms of learning combinations of skills [2]. Others have provided theoretical analysis of how transformers learn topics by studying co-occurrences of words in the training data [35]. Empirically, how models learn is still a mystery—for instance, models trained on code are found to perform fairly well at commensense reasoning [37]. Our work initiates a study on how LMs learn various skills and how to exploit this for better data selection.

**Task selection**    In multitask auxiliary learning, the goal is to train a model to perform well on a target task(s) by selecting the most beneficial source tasks to train on. One can use feature similarity to select tasks [30], but we find in our synthetics that feature similarity does not always recover skills. In Taskonomy [70], a hypergraph over a set of tasks is learned and used to select tasks. The methods used to develop the taxonomy can be applied to further expand our graph learning (e.g., studying transitive and higher-order properties). However, their focus is on task selection in computer vision rather than data selection for LMs to learn skills. Lastly, the contemporary work of TaskWeb [25] builds a graph among 22 common NLP tasks in order to determine what the best source tasks are for a target task. Their definition of an edge in the task graph is less strict than ours (their comparison is on if training on additional data from $s_i$ helps with $s_j$, while we fix the overall amount of training data over both $s_i$ and $s_j$). Overall, our approach is similar in use of the skills graph, but we incorporate it into a dynamic sampling algorithm. Furthermore, we look more broadly at skills, rather than tasks, and characterize when we expect using the skills graph to improve model performance.

**Education**    The notion of skill has been studied in education. Classical research on learning hierarchies [66] identify sets of skills that make up subordinate capabilities for students. For instance, [13] identified that in order for students to solve linear equations, there were many prerequisite

skills, ranging from the simplest being symbol recognition to the most complex being the ability to add, subtract, multiple, and divide from both sides of the equation. More recently, decision-making over lesson sequences based on skills, e.g., what the student already knows versus what the lesson teaches, has become an area of interest in personalized learning [49].

## C  Additional Algorithmic Details

### C.1  Derivation of SKILL-IT Update Rule

First, we provide the derivation of our update rule from online mirror descent using the proximal point view [19]. We restate our optimization problem from (3):

$$\underset{p_1,\ldots,p_T \in \Delta^{k-1}}{\text{minimize}} \quad \frac{1}{m}\sum_{j=1}^{m} L_{\text{eval},j}(f_T) \tag{5}$$

$$\text{s.t } L_{\text{eval},j}(f_t) = L_{\text{eval},j}(f_{t-1})(1-\alpha A_{:,j}^{\top} p_{t-1}) \, \forall j \in [m], t=1,\ldots,T$$

$$f_t = \Phi(f_{t-1}, p_{t-1}) \, \forall t = 1,\ldots T$$

Let $\bar{L}_t(p) = \frac{1}{m}\sum_{i=j}^{m} L_{\text{eval},j}(f_{t+1}) = \frac{1}{m}\sum_{i=j}^{m} L_{\text{eval},j}(\Phi(f_t,p))$; that is, $p$ is the mixture we must choose at time $t$ and $\bar{L}_t$ is the average loss per skill of the model after it is trained on $p$ at round $t$. A greedy approximation of (5) is $\underset{p\in\Delta^{k-1}}{\text{minimize}} \bar{L}_t(p)$, given the model and mixtures at previous rounds. A linear approximation of $\bar{L}_t(p)$ is

$$\bar{L}_t(p) \approx \bar{L}_t(p_{t-1}) + \langle \nabla \bar{L}_{t-1}(p_{t-1}), p - p_{t-1} \rangle \tag{6}$$

Then, the problem of minimizing $\bar{L}_t(p)$ becomes

$$\text{argmin}_{p\in\Delta^{k-1}} \langle \eta \nabla \bar{L}_{t-1}(p_{t-1}), p \rangle \tag{7}$$

after we drop terms from (6) that do not depend on $p$. Note that the $\eta$ is a constant and does not impact the solution. The optimal solution to this problem is selecting the $p$ that has the most weight on the slice with the largest gradient. To improve stability and prevent overfitting, we introduce regularization via a Bregman divergence $D_h(p||p_{t-1}) = h(p) - h(p_{t-1}) - \langle \nabla h(p_{t-1}), p - p_{t-1} \rangle$. After dropping terms that do not contain $p$, our problem is now

$$\text{argmin}_{p\in\Delta^{k-1}} \langle \eta \nabla \bar{L}_{t-1}(p_{t-1}), p \rangle + h(p) - \langle \nabla h(p_{t-1}), p \rangle \tag{8}$$

Taking the gradient and setting it equal to $0$ gives us

$$\eta \nabla \bar{L}_{t-1}(p_{t-1}) + \nabla h(p) - \nabla h(p_{t-1}) = 0 \tag{9}$$

Similar to in standard multiplicative weights, we set $h(p) = \sum_i p_i \ln p_i$ and $\nabla h(p) = [\ln p_i + 1]_i$. Then,

$$\ln p^i = \ln p_{t-1}^i - \eta \nabla_i L_{t-1}(p_{t-1})$$
$$\Rightarrow p_{t+1}^i = p_t^i \exp(-\eta \nabla_i \bar{L}_t(p_t)) \tag{10}$$

where $\nabla_i$ is the $i$th element of the gradient. Now we wish to compute $\nabla_i \bar{L}_t(p_t) = \frac{1}{m}\sum_{j=1}^{m} \nabla_i[L_{\text{eval},j}(f_{t+1})] = \frac{1}{m}\sum_{j=1}^{m} \nabla_i[L_{\text{eval},j}(\Phi(f_t,p_t))]$. Recall the dynamics model for $L_{\text{eval}}$:

$$L_{\text{eval},j}(f_{t+1}) = L_{\text{eval},j}(f_t)(1 - A_{:,j}^{\top} p_t), \tag{11}$$

The gradient of this model with respect to each training skill $s_i$ is

$$\nabla_i L_{\text{eval},j}(f_{t+1}) = -A_{ij} L_{\text{eval},j}(f_t) \tag{12}$$

$$\Rightarrow \nabla_i \bar{L}_t(p_t) = \frac{1}{m}\sum_{j=1}^{m} -A_{ij} L_{\text{eval},j}(f_t)$$

Plugging this back into (10),

$$p_{t+1}^i = p_t^i \exp\left(\eta \sum_{j=1}^{m} A_{ij} L_{\text{eval},j}(f_t)\right), \tag{13}$$

where we can absorb the $\frac{1}{m}$ into $\eta$.

---

**Algorithm 2** Brute-Force Graph Learning
---
1: **Input:** Ordered skill set $\mathcal{S} = \{s_1,...,s_k\}$. Number of training steps $K$, base model $f_0$.
2: **for** $j \in [k]$ **do**
3:     Train $f_0$ on samples from $\mathcal{X}_{s_j}$ for $K$ steps and denote $f_{K,j}$ to be the model after training.
4:     Observe change in loss, $\delta_j^j = L_{\text{eval},j}(f_0) - L_{\text{eval},j}(f_{K,j})$.
5: **end for**
6: **for** $i,j \in [k]$ **do**
7:     Train $f_0$ on samples from $\mathcal{X}_{s_i} \cup \mathcal{X}_{s_j}$ for $K$ steps and denote $f_{K,i,j}$ to be the model after training.
8:     Observe change in loss, $\delta_j^{i,j} = L_{\text{eval},j}(f_0) - L_{\text{eval},j}(f_{K,i,j})$.
9:     **if** $\delta_j^{ij} > \delta_j^j$ **then**
10:         Draw edge $s_i \rightarrow s_j$ and set $A_{ij} > 0$.
11:     **end if**
12: **end for**
13: **return** Adjacency Matrix $A \in \mathbb{R}^{k \times k}$

---

**Algorithm 3** Approximate Graph Learning
---
1: **Input:** Ordered skill sets $\mathcal{S}_{\text{train}}$ and $\mathcal{S}_{\text{eval}}$. Number of training steps $K$, base model $f_0$.
2: **for** $i \in [k]$ **do**
3:     Train $f_0$ on samples from $\mathcal{X}_{s_{\text{train},i}}$ for $K$ steps and denote $f_{K,i}$ to be the model after training.
4:     **for** $j \in [m]$ **do**
5:         Observe change in loss, $\delta_j^i = L_{\text{eval},j}(f_0) - L_{\text{eval},j}(f_{K,i})$.
6:         If $\delta_j^i > 0$, draw edge $s_{\text{train},i} \rightarrow s_{\text{train},j}$ and set $A_{ij} > 0$.
7:     **end for**
8: **end for**
9: **return** Bipartite Adjacency Matrix $A \in \mathbb{R}^{k \times m}$

---

### C.2 Graph Learning Method

We provide algorithms for learning the graph over an ordered skill set. In Algorithm 2, we discuss the brute-force approach for learning the adjacency matrix. This approach only works when $\mathcal{S}_{\text{eval}} \subseteq \mathcal{S}_{\text{train}}$ (e.g. pre-training and fine-tuning cases), so we denote $\mathcal{S} = \mathcal{S}_{\text{train}}$ in the algorithm box. In Algorithm 3, we discuss the linear approach for learning the adjacency matrix. This approach works even in the out-of-domain case when $\mathcal{S}_{\text{eval}}$ and $\mathcal{S}_{\text{train}}$ are disjoint.

In both approaches, the exact value of $A_{ij}$ can vary, but we can typically set it proportional to $\delta_j^{i,j} - \delta_j^j$ in the brute-force case or $\delta_j^i$ in the approximate case. The exact constructions and methods for learning each $A$ in our experiments are in Appendix D.2.

## D Additional Experimental Details

### D.1 Datasets

We present details about each dataset used, including information on the skills and the validation dataset. A summary is presented in Table 2.

- Alpaca dataset [56]: the Alpaca dataset consists of 52K instruction examples that were generated from text-davinci-003. We applied the Berkeley Neural Parser [27, 28] to each instruction, keeping 40777 samples it was able to parse successfully. If the sample began with a question, we annotated it with the skill "question", and otherwise we annotated it with the verb identified from the parser. We grouped the data into a total of 38 skills, such as "list", "edit", "calculate", "describe" and "identify".

- Pile of Law [22]: the Pile of Law dataset consists of various sources of legal and administrative data, ranging from tax rulings to the world's constitutions. We evaluate on a subset of the Pile of Law validation dataset consisting of 13883 samples, where we selected max(645, source size) samples per source. We truncated each sample to be no more than 100K characters.

- LEGO [72]: for the LEGO synthetic, we set $k = 5$ and sample 192000 points across the skills. Our validation dataset consisted of 100 samples per skill.

Table 2: We list each dataset used as well as its corresponding skill. We include the number of skills in the training dataset, as well as details on how the validation dataset is constructed.

| Dataset | Skill | # skills | Validation data |
| --- | --- | --- | --- |
| Alpaca | Instruction type | 38 | 50 samples per skill |
| Pile of Law | Legal data source | 31 | 645 samples per skill |
| LEGO | Reasoning chain depth | 5 | 100 samples per skill |
| Addition | Digit | 3 | 100 samples per skill |
| NI (pre-training) | Task category | 23 | 50 samples per task |
| NI (Spanish QG) | Task category $\times$ language | 4 | 100 samples per task |
| NI (stance detection) | Task category | 2 | 50 samples per task |
| NI (out-of-domain) | Task category | 59,12 | 400 samples per task |
| RedPajama | Data source | 7 | LM eval harness |

- Addition: for the 3-digit addition synthetic, we set $k=3$ and sample 192000 points across the skills. We use a validation dataset of 100 samples per skill.
- Natural Instructions [63, 41]: the Natural Instructions dataset is a large collection of tasks and their definitions in natural language. For the pre-training setting, we used a set of 23 task categories that had the largest degree (in-degree + out-degree) in the learned skills graph, for a total of 1,232,437 samples and 425 tasks to select from. We evaluated on 50 samples per task.

  For the fine-tuning setting with Spanish question generation, we select data over 4 skills (Spanish question generation, Spanish question answering, English question generation, English question answering) for a total of 513210 samples and 212 tasks to select from. We evaluated on 100 samples per task.

  For the fine-tuning setting with stance detection, we select data over 2 skills (stance detection, text matching) for a total of 50990 samples and 19 tasks to select from. We evaluated on 50 samples per task.

  For the out-of-domain setting, we select data over all 59 task categories for a total of 2,417,867 samples and 753 tasks to select from. The test split consisted of 12 task categories and 119 tasks, and we evaluated on min(400, task size) samples per task.
- RedPajama [57]: the RedPajama dataset is a 1-trillion token dataset that aims to reproduce the LLaMA [59] training dataset. We select over the 7 data sources and evaluate using the LM evaluation harness [15].

### D.2 Graph Learning Details

We describe how the skills graph was learned on each dataset.

- Alpaca (Figure 9): we use Algorithm 3 and train for $K=150$ steps per skill. Each edge $i \to j$ has a weight of $\delta_j^i$, the difference in loss on skill $j$ before and after training on $i$. Next, we compare the average validation loss of skill-stratified sampling versus random sampling when we train for $K=1000$ steps. We find that skill-stratified sampling only does 0.007 better than random sampling, confirming that Alpaca's dense skills graph suggests that random sampling is the best we can do.
- Pile of Law (Figure 10): we use Algorithm 3 and train for $K=150$ steps. Each edge $i \to j$ has a weight of $\delta_j^i$, the difference in loss on skill $j$ before and after training on $i$.
- LEGO (Figure 11): we use both Algorithm 2 and Algorithm 3 and train for $K=6000$ steps each. Each edge $i \to j$ has a weight of 0.5 if the amount of data associated with skill $j$ that is needed to reach 0.01 validation loss is less when training on $(i,j)$ than on $j$ (edges are set to 0 if 0.01 validation loss is not reached, even if loss is decreasing). Each edge $i \to j$ is also set to 0.5 if training on $i$ decreases loss directly on $j$. We set each diagonal entry of $A$ to be 1.
- Addition (Figure 12): we use Algorithm 2 and train for $K=6000$ steps. Each edge $i \to j$ has a weight of 0.5 if the amount of data associated with skill $j$ that is needed to reach 0.01 validation loss is less when training on $(i,j)$ than on $j$ (edges are set to 0 if 0.01 validation loss is not reached, even if loss is decreasing). We set each diagonal entry of $A$ to be 1.
- Natural Instructions (Figure 13, 14, 15): we use Algorithm 3. For the pre-training setting, we train for $K=600$ steps and assign each edge $i \to j$ a weight $\delta_j^i$ equal to the change in loss on $j$ in the first 100 steps for all $i,j \in [k]$, including diagonal entries. For the fine-tuning setting, we train for $K=600$ steps and assign each edge $i \to j$ a weight $\delta_j^i$ equal to the change in loss before and after training. For the out-of-domain setting, we train for $K=600$ steps and assign each edge $i \to j$ a weight $\delta_j^i$ equal to the change in loss before and after training in the first 100 steps.

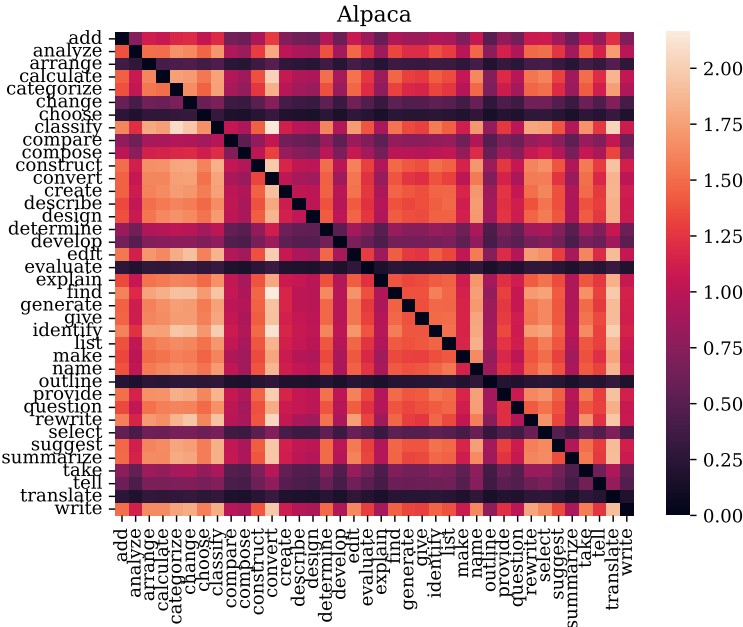

Figure 9: Alpaca heatmap where $i,j$th entry is $\max(0,\delta_j^i)$ (the change in loss on $s_j$ after training on $s_i$ for 150 steps). Diagonal entries are set to 0 for clearer visualization.

- RedPajama (Figure 16): we use Algorithm 3 and train for 1 billion tokens per data source. We assign each edge $i \rightarrow j$ a weight $\delta_j^i$ equal to the change in perplexity on the validation datalsoa before and after training.

## D.3  Training Details

We describe the parameters used for SKILL-IT.

### SKILL-IT pre-training

- LEGO: $\eta = 0.5, T = 6, w = 3$. We train for 6000 steps.
- Addition: $\eta = 0.1, T = 5, w = 3$. We train for 6000 steps.
- Natural Instructions (pre-training): $\eta = 0.2, T = 1$. We train for 5000 steps.

For the LEGO random baseline, when we selected points at random, we used an imbalanced training dataset with proportions 1:1:1:3:5. For the addition random baseline, we used an imbalanced dataset with randomly selected proportions: 13:14:18. For the curriculum learning baselines, the pacing function, $g(i)$, denotes the size of the subset of the highest scoring samples that we uniformly select from in the $i$th epoch. We define our pacing function as $g(i) = \frac{iH}{M}$, where $H$ is the number of steps and $M$ is 5 epochs for LEGO and NI, and 3 for addition.

### SKILL-IT fine-tuning

- LEGO: $\eta = 0.5, T = 10, w = 3$. We train for 6000 steps.
- Addition: $\eta = 0.1, T = 5, w = 3$. We train for 6000 steps.
- Natural Instructions (Spanish QG): $\eta = 0.8, T = 6, w = 3$. We train for 600 steps.
- Natural Instructions (stance detection): $\eta = 0.2, T = 6, w = 3$. We train for 600 steps.

### SKILL-IT out-of-domain

- Natural Instructions: $\eta = 0.2, T = 10, w = 3$. We train for 5000 steps.
- RedPajama: $\eta = 100, T = 1$. We train for 3 billion tokens.

All results are computed over 5 random seeds.

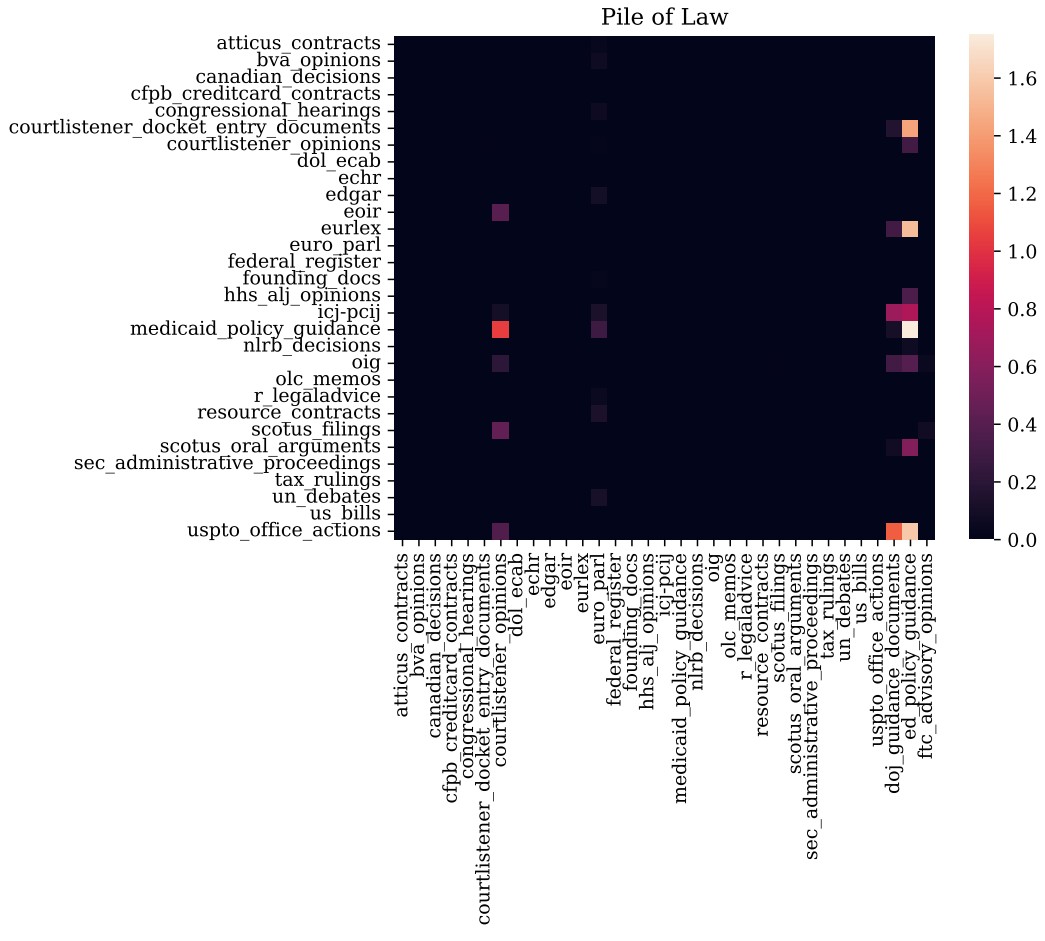

Figure 10: Pile of Law heatmap where $i,j$th entry is $\max(0,\delta_j^i)$ (the change in loss on $s_j$ after training on $s_i$ for 150 steps). Diagonal entries are set to 0 for clearer visualization.

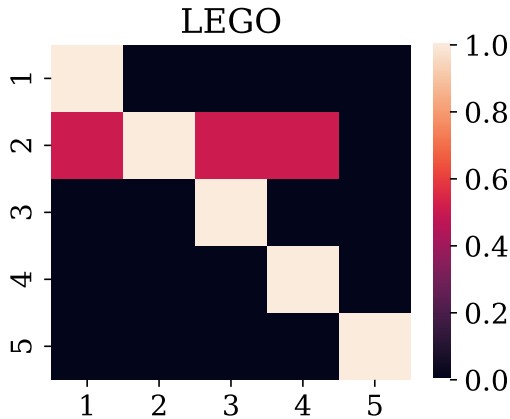

Figure 11: LEGO heatmap with $k=5$ where $i,j$th entry is set to $0.5$ if the number of steps needed to reach $0.01$ loss on skill $j$ when training on a balanced mixture of skills $i$ and $j$ is less than when training on skill $j$ only.

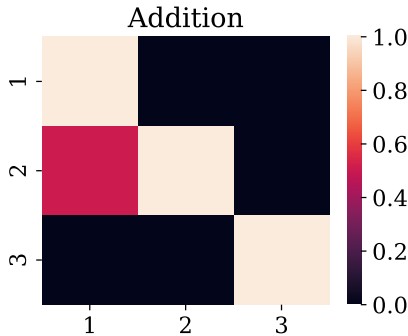

Figure 12: Addition heatmap with $k=3$ where $i,j$th entry is set to $0.5$ if the number of steps needed to reach $0.01$ loss on skill $j$ when training on a balanced mixture of skills $i$ and $j$ is less than when training on skill $j$ only.

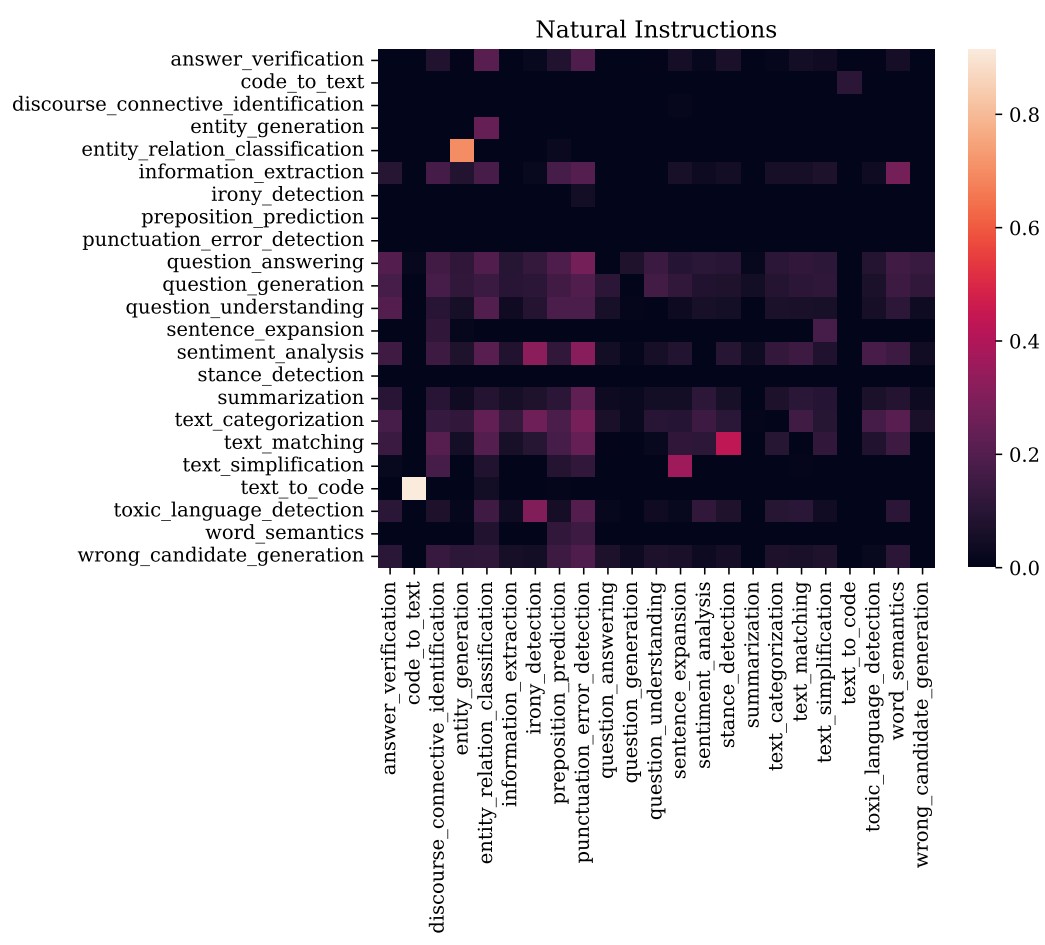

Figure 13: Natural Instructions heatmap where $i,j$th entry is $\max(0,\delta_j^i)$ (the change in loss on $s_j$ after training on $s_i$ for 100 steps). Diagonal entries are set to 0 for clearer visualization.

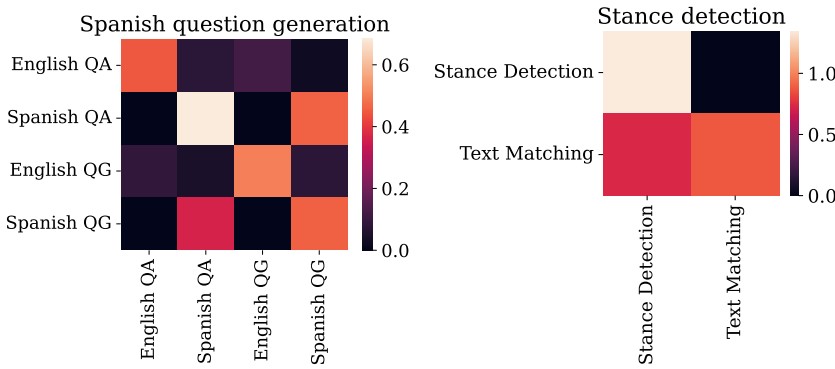

Figure 14: Spanish question generation and stance detection heatmaps where $i,j$th entry is $\max(0,\delta_j^i)$ (the change in loss on $s_j$ after training on $s_i$ for 100 steps).

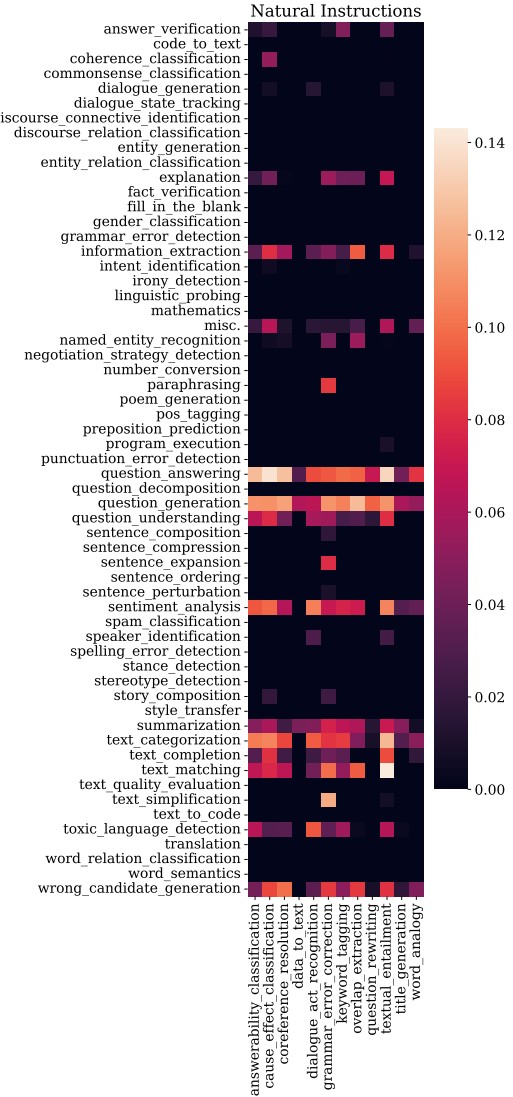

Figure 15: Natural Instructions heatmap for out-of-domain setting where rows are for the training skills and columns are for the evaluation skills. The $i,j$th entry is $\max(0,\delta_j^i)$ (the change in loss on $s_j$ after training on $s_i$ for 100 steps).

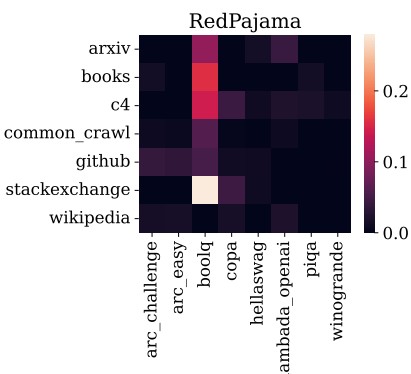

Figure 16: RedPajama heatmap for out-of-domain setting where rows are for the training skills and columns are for the evaluation skills. The $i,j$th entry is $\max(0, \delta_j^i)$ (the change in perplexity on $s_j$ after training on $s_i$ for 1B tokens).

Batch sizes of $32$ and $64$ were used for the LEGO and addition synthetic on the 125M and 1.3B parameter model, respectively. Batch sizes of $4$ and $16$ were used for the Natural Instructions experiments on the 125M and 1.3B parameter model.

For the out-of-domain Natural Instructions experiment and Alpaca graph learning experiments, a learning rate of 5e-6 with linear scheduler and $50$ warmup steps was used. All other experiments used a learning rate of 5e-5. All experiments used AdamW with betas = 0.9, 0.999, eps = 1e-8, and weight decay = 0.01. A context window of $512$ was used for all experiments except LEGO and addition, which used a window of $128$.

Experiments with the Addition dataset were run using an Nvidia RTX A6000. Other experiments using the GPT-Neo 125M parameter model were run on an Nvidia Tesla P100. Experiments using the GPT-Neo 1.3B parameter model were run on an Nvidia Tesla A100.

# E  Additional Experimental Results

## E.1  Additional examples of LEGO ordered skill sets

For the LEGO synthetic, it may appear obvious that the skills graph is equivalent to the reasoning chain over the variables. However, in Figure 17 we see that this is not the case. Training on skills 2 and 4 together results in lower loss on skill 4 than when trained on skill 4 alone. However, training on skills 3 and 4 together results in roughly the same loss on skill 4 as when training on skill 4 alone, even though skill 3 and skill 4 share an edge in the LEGO synthetic's underlying reasoning chain. This suggests that our intuition for how skills influence each other does not always match how the model learns skills.

Next, we consider a slightly more complex reasoning pattern on the LEGO synthetic. Instead of a chain, we construct a tree, where two variables in the LEGO synthetic are both defined in terms of the same parent variable. For example,

Input: c = val 1, y = not w, v = val c, w = not c. Output: y = 1.

In this example, $k = 4$ and both $v$ and $w$ are written in terms of $c$, and the reasoning graph has edges $1 \rightarrow 2, 1 \rightarrow 3, 2 \rightarrow 4$. In this case, we see that training on skill 2 or skill 3 both improve losses on skills 2 and 3 (Figure 18). However, unlike the previous figures, training on skills 2 and 4 or skills 3 and 4 do not significantly help reduce loss on skill 4 (Figure 19). Again, these measurements demonstrate that the reasoning graph does not necessarily equal the skills graph.

## E.2  Unsupervised skill recovery

We explore several clustering techniques for recovering the skills in the LEGO synthetic on the validation dataset. Our results are shown in Table 3.

We first cluster based on the pre-trained model embeddings of the last token and the average token. We also report accuracies of clustering based on the trained model embedding's last token, where we train

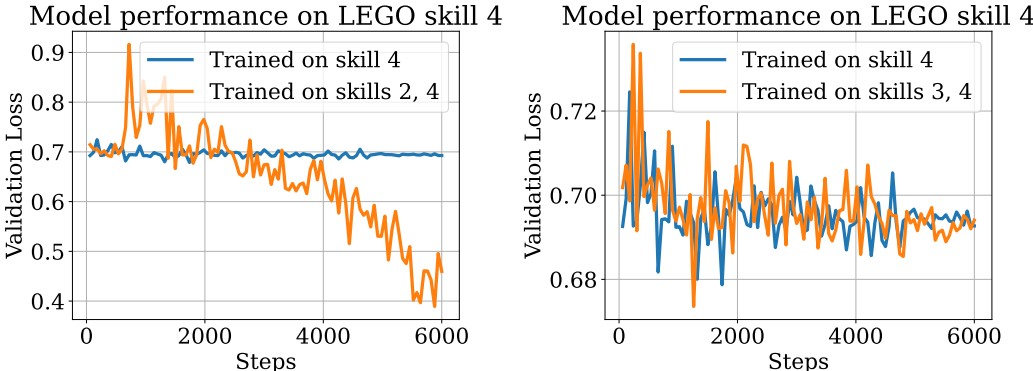

Figure 17: Performance on LEGO skill 4 when training on skill 4, skills 2 and 4, and skills 3 and 4. Even though skill 3 and skill 4 share an edge in the LEGO synthetic's underlying reasoning chain (i.e. a model predicting correct for the fourth variable is one extra step beyond predicting correct for the third variable), we find that training on skills 2 and 4 helps improve performance on skill 4 more.

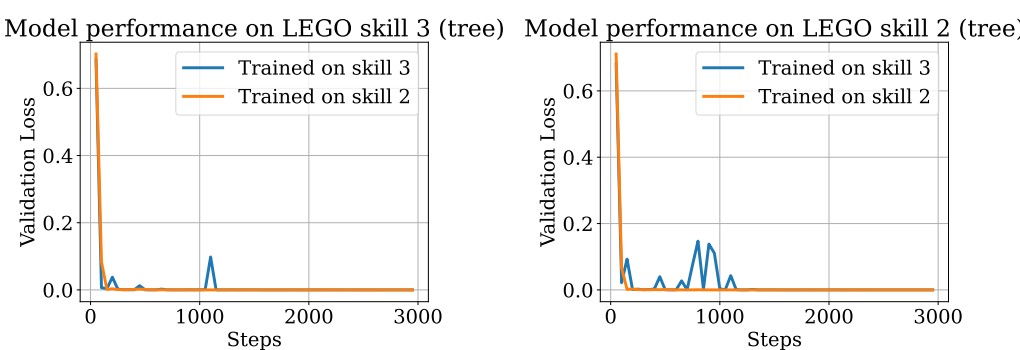

Figure 18: Performance on LEGO skill 2 and 3 when training on skills 2 and 3. The reasoning pattern is a tree rather than a chain over $k = 4$ variables. Skills 2 and 3 are at the same "depth" in the graph and both depend on skill 1, so there is positive influence between the skills despite there being no edge between 2 and 3 in the LEGO reasoning graph.

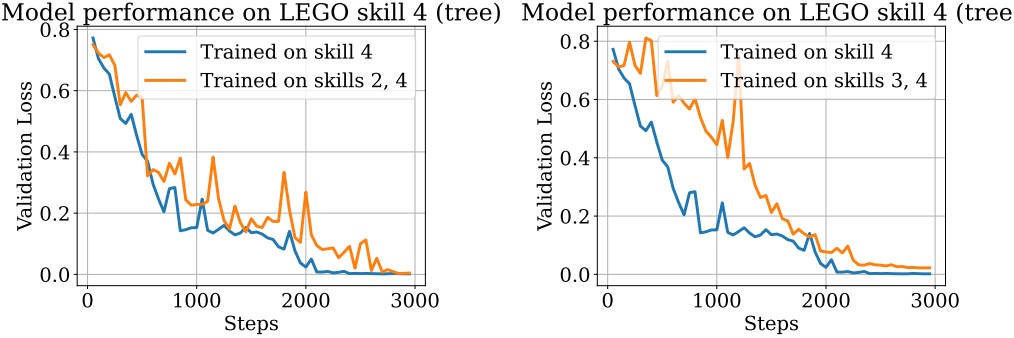

Figure 19: Performance on LEGO skill 4 when training on skills 2,4 and skills 3,4. We find that in both cases, the benefit from training on additional skills is minor. For instance, training on 2 and 4 reaches $0.01$ loss in 2700 steps, while training on 4 only reaches it in 2100 steps.

Table 3: Clustering-based skill recovery methods on the LEGO dataset. The validation dataset we cluster consists of $500$ points with $k = 5$, and results are reported over 10 runs of k-means.

| Cluster method | Accuracy |
|---|---|
| Pretrained embedding of last token | $24.8 \pm 0.5$ |
| Pretrained embedding of average token | $25.2 \pm 1.1$ |
| Trained model embedding of last token | $38.4 \pm 0.8$ |
| Sentence-BERT embedding | $23.9 \pm 0.7$ |
| Losses over multiple runs | $\mathbf{61.0 \pm 1.6}$ |

the model using random sampling for 6000 steps, and clustering based on Sentence-BERT embeddings. Among these four methods, using the trained model embeddings has the highest accuracy of 38.4 points.

Next, we cluster points based on losses. In particular, we do 10 runs, each for 6000 steps and with a randomly sampled mixture of skills. For each run, we evaluate the model on the validation dataset at 120 checkpoints. Then, each sample in the validation dataset has 1200 losses associated with it, comprising a feature vector for that sample. We perform k-means clustering on these features, which has an accuracy of 61.0 points, significantly higher than the second best accuracy of 38.4.

## E.3    Full results for Section 5

### E.3.1    Per-skill performance

In this section, we provide tables containing the per skill break-down of our results from Section 5.

**Continual Pre-training**    In the continual pre-training setting, we report two additional baselines that combine curriculum learning with skills. Curriculum learning has been proposed for multitask learning [60], in which groups of data are ranked by their average score and then trained in order of this ranking (with mixing of previously seen groups to avoid forgetting). We construct two baselines, Skill-curriculum and Skill-anticurriculum, using Algorithm 1 from [60]. In contrast to the random baseline which has imbalanced skills, this approach has knowledge of skills and thus uses a skill-stratified training dataset to sample from. We set the fraction of the previous group to be frac $= 0.4$, as we found that setting frac $= 0.0$ resulted in forgetting.

We report loss per skill for the LEGO synthetic in Table 4, which corresponds to the results in Figure 4. We report accuracy per skill in Table 5 and Figure 20. We report the loss per skill for the Addition synthetic in Table 6, which also correspond to to the results in Figure 4, and we provide the accuracy per skill in Figure 21 (we omit a table of accuracies since most methods attained $100\%$ accuracy by the end of training). Finally, we report validation loss per task category for the Natural Instructions continual pre-training experiment in Table 7, where we find that SKILL-IT outperforms random sampling by $3.2\%$ on average across skills.

Table 4: Results on validation loss per skill for LEGO pre-training experiment, averaged over 5 random seeds.

| | Skill 1 | Skill 2 | Skill 3 | Skill 4 | Skill 5 | Average |
|---|---|---|---|---|---|---|
| Random | $0_{\pm 0.000}$ | $0.669_{\pm 0.051}$ | $0.686_{\pm 0.013}$ | $0.676_{\pm 0.046}$ | $0.678_{\pm 0.032}$ | $0.542_{\pm 0.028}$ |
| Curriculum | $0.003_{\pm 0.002}$ | $0.431_{\pm 0.028}$ | $0.875_{\pm 0.051}$ | $0.140_{\pm 0.022}$ | $0.001_{\pm 0.000}$ | $0.290_{\pm 0.008}$ |
| Anticurriculum | $0.047_{\pm 0.012}$ | $0.351_{\pm 0.011}$ | $0.601_{\pm 0.128}$ | $0.121_{\pm 0.017}$ | $0.003_{\pm 0.001}$ | $0.225_{\pm 0.026}$ |
| Skill-stratified | $0_{\pm 0.000}$ | $0.045_{\pm 0.036}$ | $0.056_{\pm 0.029}$ | $0.080_{\pm 0.044}$ | $0.050_{\pm 0.025}$ | $0.046_{\pm 0.022}$ |
| Skill-curriculum | $0_{\pm 0.000}$ | $0.639_{\pm 0.101}$ | $0.701_{\pm 0.012}$ | $0.697_{\pm 0.002}$ | $0.693_{\pm 0.002}$ | $0.546_{\pm 0.018}$ |
| Skill-anticurriculum | $0.001_{\pm 0.001}$ | $0.306_{\pm 0.218}$ | $0.541_{\pm 0.252}$ | $0.596_{\pm 0.201}$ | $0.611_{\pm 0.172}$ | $0.411_{\pm 0.144}$ |
| SKILL-IT | $0.004_{\pm 0.005}$ | $0.009_{\pm 0.011}$ | $0.029_{\pm 0.016}$ | $0.043_{\pm 0.020}$ | $0.019_{\pm 0.006}$ | $0.021_{\pm 0.009}$ |

Table 5: Results on accuracy per skill for LEGO pre-training experiment, averaged over 5 random seeds.

|  | Skill 1 | Skill 2 | Skill 3 | Skill 4 | Skill 5 | Average |
|---|---|---|---|---|---|---|
| Random | $100.0_{\pm 0.0}$ | $53.8_{\pm 6.8}$ | $57.4_{\pm 5.5}$ | $48.2_{\pm 6.1}$ | $51.2_{\pm 4.3}$ | $62.1_{\pm 3.5}$ |
| Curriculum | $100.0_{\pm 0.0}$ | $80.2_{\pm 1.3}$ | $49.6_{\pm 3.0}$ | $97.0_{\pm 2.3}$ | $100.0_{\pm 0.0}$ | $85.4_{\pm 0.7}$ |
| Anticurriculum | $98.8_{\pm 1.3}$ | $84.6_{\pm 3.0}$ | $68.0_{\pm 6.9}$ | $96.8_{\pm 1.3}$ | $100.0_{\pm 0.0}$ | $89.6_{\pm 1.9}$ |
| Skill-stratified | $100.0_{\pm 0.0}$ | $98.2_{\pm 1.8}$ | $98.2_{\pm 1.3}$ | $97.8_{\pm 1.6}$ | $98.2_{\pm 1.3}$ | $98.5_{\pm 0.9}$ |
| Skill-curriculum | $100.0_{\pm 0.0}$ | $57.4_{\pm 15.7}$ | $50.4_{\pm 3.6}$ | $46.8_{\pm 7.4}$ | $51.8_{\pm 3.7}$ | $61.3_{\pm 4.8}$ |
| Skill-anticurriculum | $100.0_{\pm 0.0}$ | $86.8_{\pm 19.8}$ | $65.4_{\pm 19.7}$ | $61.2_{\pm 16.6}$ | $62.0_{\pm 13.4}$ | $75.1_{\pm 11.5}$ |
| SKILL-IT | $99.8_{\pm 0.4}$ | $99.6_{\pm 0.9}$ | $99.0_{\pm 0.7}$ | $98.6_{\pm 0.9}$ | $99.6_{\pm 0.9}$ | $99.3_{\pm 0.4}$ |

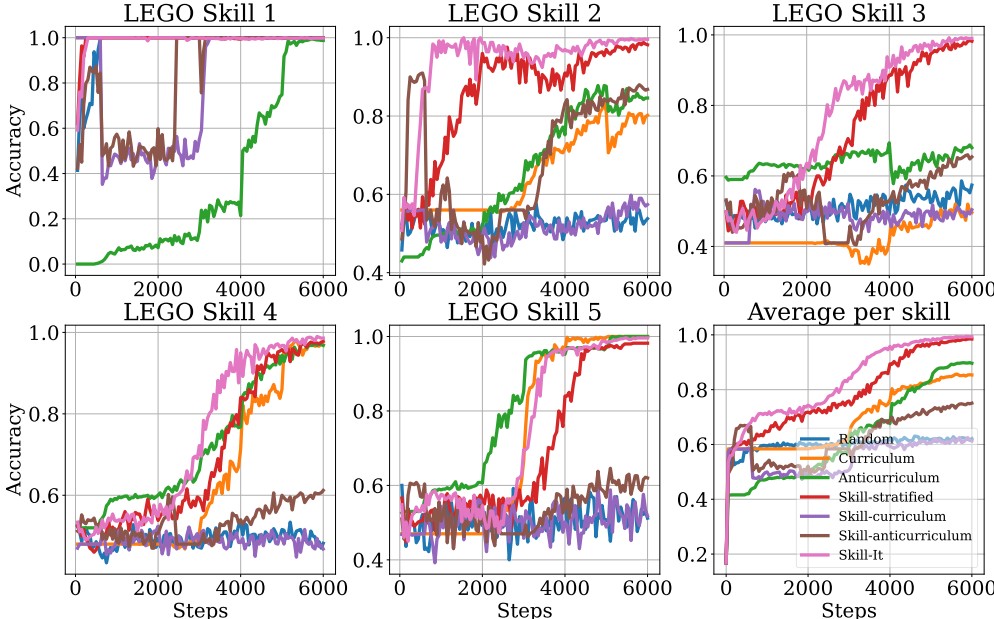

Figure 20: Accuracy of SKILL-IT on each skill on the LEGO synthetic (if the desired variable has the correct value out of $0$ or $1$) in the continual pre-training setting. SKILL-IT attains higher accuracy more quickly than baselines that both do and do not utilize the notion of skills.

Table 6: Results on validation loss per skill for Addition pre-training experiment, averaged over 5 random seeds.

|  | Skill 1 | Skill 2 | Skill 3 | Average |
|---|---|---|---|---|
| Random | $0.008_{\pm 0.006}$ | $0.016_{\pm 0.004}$ | $0.006_{\pm 0.001}$ | $0.010_{\pm 0.003}$ |
| Curriculum | $0.009_{\pm 0.005}$ | $0.024_{\pm 0.006}$ | $0.018_{\pm 0.007}$ | $0.017_{\pm 0.006}$ |
| Anticurriculum | $0.004_{\pm 0.001}$ | $0.011_{\pm 0.002}$ | $0.016_{\pm 0.003}$ | $0.011_{\pm 0.002}$ |
| Skill-stratified | $0.015_{\pm 0.012}$ | $0.017_{\pm 0.004}$ | $0.010_{\pm 0.004}$ | $0.014_{\pm 0.006}$ |
| Skill-curriculum | $0.027_{\pm 0.046}$ | $0.700_{\pm 1.018}$ | $0.661_{\pm 0.884}$ | $0.463_{\pm 0.630}$ |
| Skill-anticurriculum | $0.478_{\pm 1.037}$ | $0.194_{\pm 0.269}$ | $0.188_{\pm 0.256}$ | $0.287_{\pm 0.451}$ |
| SKILL-IT | $0.003_{\pm 0.001}$ | $0.008_{\pm 0.003}$ | $0.009_{\pm 0.005}$ | $0.007_{\pm 0.002}$ |

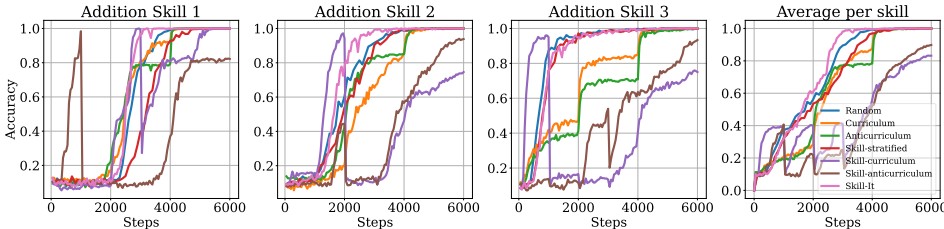

Figure 21: Accuracy of SKILL-IT on each skill on the addition synthetic (10-way accuracy if the predicted digit is correct) in the continual-pretraining setting. On average, SKILL-IT attains higher accuracy more quickly than baselines that both do and do not utilize the notion of skills.

Table 7: Validation loss per skill for data selection in continual pre-training setting on a subset of the Natural Instructions Dataset.

| Skill | Random | Curriculum | Anticurriculum | Skill-stratified | Skill-curriculum | Skill-anticurriculum | Skill-It |
|---|---|---|---|---|---|---|---|
| Answer Verification | $2.297_{\pm 0.058}$ | $2.368_{\pm 0.055}$ | $2.391_{\pm 0.061}$ | $2.180_{\pm 0.059}$ | $2.249_{\pm 0.116}$ | $2.325_{\pm 0.085}$ | $2.158_{\pm 0.059}$ |
| Code to Text | $0.246_{\pm 0.021}$ | $0.203_{\pm 0.019}$ | $1.099_{\pm 0.115}$ | $0.178_{\pm 0.016}$ | $0.126_{\pm 0.009}$ | $1.232_{\pm 0.070}$ | $0.223_{\pm 0.017}$ |
| Discourse Connective Identification | $2.927_{\pm 0.069}$ | $3.084_{\pm 0.067}$ | $2.932_{\pm 0.058}$ | $2.805_{\pm 0.071}$ | $2.891_{\pm 0.001}$ | $2.925_{\pm 0.011}$ | $2.784_{\pm 0.068}$ |
| Entity Generation | $2.033_{\pm 0.421}$ | $2.012_{\pm 0.437}$ | $2.363_{\pm 0.234}$ | $1.803_{\pm 0.384}$ | $1.853_{\pm 0.483}$ | $2.068_{\pm 0.719}$ | $1.863_{\pm 0.418}$ |
| Entity Relation Classification | $1.020_{\pm 0.147}$ | $1.014_{\pm 0.140}$ | $1.533_{\pm 0.138}$ | $0.859_{\pm 0.131}$ | $0.825_{\pm 0.022}$ | $0.959_{\pm 0.009}$ | $0.908_{\pm 0.146}$ |
| Information Extraction | $2.154_{\pm 0.040}$ | $2.247_{\pm 0.037}$ | $2.352_{\pm 0.042}$ | $2.140_{\pm 0.037}$ | $2.286_{\pm 0.022}$ | $2.338_{\pm 0.025}$ | $2.073_{\pm 0.042}$ |
| Irony Detection | $3.024_{\pm 0.154}$ | $3.798_{\pm 0.095}$ | $2.942_{\pm 0.158}$ | $2.680_{\pm 0.146}$ | $3.889_{\pm 0.066}$ | $2.099_{\pm 0.152}$ | $2.797_{\pm 0.155}$ |
| Preposition Prediction | $0.979_{\pm 0.124}$ | $0.887_{\pm 0.147}$ | $1.488_{\pm 0.213}$ | $0.845_{\pm 0.152}$ | $0.941_{\pm 0.019}$ | $1.044_{\pm 0.029}$ | $0.876_{\pm 0.173}$ |
| Punctuation Error Detection | $2.950_{\pm 0.065}$ | $3.120_{\pm 0.052}$ | $2.961_{\pm 0.064}$ | $3.264_{\pm 0.061}$ | $3.019_{\pm 0.010}$ | $3.360_{\pm 0.013}$ | $3.216_{\pm 0.055}$ |
| Question Answering | $2.277_{\pm 0.005}$ | $2.367_{\pm 0.006}$ | $2.398_{\pm 0.006}$ | $2.542_{\pm 0.004}$ | $2.689_{\pm 0.001}$ | $2.707_{\pm 0.016}$ | $2.448_{\pm 0.008}$ |
| Question Generation | $2.617_{\pm 0.005}$ | $2.777_{\pm 0.015}$ | $2.695_{\pm 0.008}$ | $2.783_{\pm 0.021}$ | $3.062_{\pm 0.006}$ | $2.876_{\pm 0.032}$ | $2.666_{\pm 0.012}$ |
| Question Understanding | $1.965_{\pm 0.051}$ | $2.199_{\pm 0.059}$ | $2.060_{\pm 0.033}$ | $1.958_{\pm 0.051}$ | $2.385_{\pm 0.022}$ | $2.100_{\pm 0.054}$ | $1.895_{\pm 0.043}$ |
| Sentence Expansion | $2.501_{\pm 0.095}$ | $2.598_{\pm 0.097}$ | $2.583_{\pm 0.074}$ | $2.225_{\pm 0.095}$ | $2.311_{\pm 0.076}$ | $2.408_{\pm 0.074}$ | $2.236_{\pm 0.083}$ |
| Sentiment Analysis | $3.203_{\pm 0.012}$ | $3.415_{\pm 0.016}$ | $3.209_{\pm 0.010}$ | $3.278_{\pm 0.014}$ | $3.607_{\pm 0.012}$ | $3.308_{\pm 0.015}$ | $3.213_{\pm 0.012}$ |
| Stance Detection | $1.810_{\pm 0.100}$ | $1.775_{\pm 0.120}$ | $2.231_{\pm 0.128}$ | $1.385_{\pm 0.070}$ | $1.361_{\pm 0.114}$ | $1.823_{\pm 0.189}$ | $1.556_{\pm 0.125}$ |
| Summarization | $2.961_{\pm 0.015}$ | $3.149_{\pm 0.023}$ | $3.041_{\pm 0.014}$ | $2.960_{\pm 0.019}$ | $3.323_{\pm 0.028}$ | $3.021_{\pm 0.013}$ | $2.907_{\pm 0.012}$ |
| Text Categorization | $2.488_{\pm 0.023}$ | $2.692_{\pm 0.029}$ | $2.553_{\pm 0.006}$ | $2.570_{\pm 0.015}$ | $3.001_{\pm 0.007}$ | $2.635_{\pm 0.014}$ | $2.448_{\pm 0.017}$ |
| Text Matching | $2.177_{\pm 0.059}$ | $2.232_{\pm 0.055}$ | $2.316_{\pm 0.048}$ | $2.152_{\pm 0.061}$ | $2.324_{\pm 0.004}$ | $2.304_{\pm 0.035}$ | $2.093_{\pm 0.054}$ |
| Text Simplification | $2.155_{\pm 0.023}$ | $2.193_{\pm 0.039}$ | $2.325_{\pm 0.033}$ | $1.926_{\pm 0.026}$ | $2.037_{\pm 0.005}$ | $2.156_{\pm 0.011}$ | $1.952_{\pm 0.026}$ |
| Text to Code | $0.560_{\pm 0.037}$ | $0.495_{\pm 0.036}$ | $1.215_{\pm 0.052}$ | $0.490_{\pm 0.029}$ | $0.433_{\pm 0.014}$ | $1.455_{\pm 0.086}$ | $0.553_{\pm 0.042}$ |
| Toxic Language Detection | $3.106_{\pm 0.027}$ | $3.496_{\pm 0.017}$ | $3.058_{\pm 0.029}$ | $3.199_{\pm 0.024}$ | $3.758_{\pm 0.025}$ | $3.155_{\pm 0.050}$ | $3.129_{\pm 0.020}$ |
| Word Semantics | $2.092_{\pm 0.027}$ | $2.334_{\pm 0.034}$ | $2.156_{\pm 0.064}$ | $1.916_{\pm 0.043}$ | $1.784_{\pm 0.048}$ | $2.424_{\pm 0.038}$ | $1.952_{\pm 0.019}$ |
| Wrong Candidate Generation | $2.438_{\pm 0.021}$ | $2.606_{\pm 0.039}$ | $2.519_{\pm 0.027}$ | $2.506_{\pm 0.026}$ | $2.849_{\pm 0.029}$ | $2.574_{\pm 0.018}$ | $2.432_{\pm 0.025}$ |
| Average | $2.173_{\pm 0.028}$ | $2.307_{\pm 0.025}$ | $2.366_{\pm 0.026}$ | $2.115_{\pm 0.027}$ | $2.304_{\pm 0.031}$ | $2.317_{\pm 0.052}$ | $2.103_{\pm 0.032}$ |

**Out-of-domain** In Table 8, we provide a breakdown of validation loss per evaluation skill under random sampling on the training data, skill-stratified sampling over prerequisite skills (e.g., the nonzero rows in Figure 15), and SKILL-IT.

Table 8: Validation loss per skill for data selection in out-of-domain setting over Natural Instructions train task split and test task split.

| Skill | Random | Skill-stratified | SKILL-IT |
|---|---|---|---|
| Answerability Classification | $3.048_{\pm 0.003}$ | $3.076_{\pm 0.002}$ | $3.043_{\pm 0.003}$ |
| Cause Effect Classification | $2.068_{\pm 0.004}$ | $2.101_{\pm 0.005}$ | $2.067_{\pm 0.006}$ |
| Coreference Resolution | $3.101_{\pm 0.003}$ | $3.142_{\pm 0.004}$ | $3.099_{\pm 0.004}$ |
| Data to Text | $2.363_{\pm 0.004}$ | $2.388_{\pm 0.005}$ | $2.359_{\pm 0.005}$ |
| Dialogue Act Recognition | $2.329_{\pm 0.009}$ | $2.364_{\pm 0.010}$ | $2.320_{\pm 0.009}$ |
| Grammar Error Correction | $2.399_{\pm 0.008}$ | $2.418_{\pm 0.009}$ | $2.389_{\pm 0.007}$ |
| Keyword Tagging | $2.744_{\pm 0.005}$ | $2.760_{\pm 0.007}$ | $2.733_{\pm 0.006}$ |
| Overlap Extraction | $2.749_{\pm 0.011}$ | $2.763_{\pm 0.012}$ | $2.733_{\pm 0.010}$ |
| Question Rewriting | $2.591_{\pm 0.009}$ | $2.628_{\pm 0.011}$ | $2.586_{\pm 0.010}$ |
| Textual Entailment | $2.472_{\pm 0.002}$ | $2.503_{\pm 0.003}$ | $2.468_{\pm 0.002}$ |
| Title Generation | $3.027_{\pm 0.002}$ | $3.037_{\pm 0.002}$ | $3.015_{\pm 0.002}$ |
| Word Analogy | $1.665_{\pm 0.016}$ | $1.682_{\pm 0.015}$ | $1.668_{\pm 0.016}$ |
| Average | $2.546_{\pm 0.003}$ | $2.572_{\pm 0.003}$ | $2.540_{\pm 0.003}$ |

In Table 9 we provide a breakdown of the RedPajama experiment's accuracy per evaluation skill, corresponding to the results in Figure 8.

Table 9: Performance of model trained on RedPajama with uniform sampling and SKILL-IT on LM evaluation harness. Unless otherwise noted, accuracy is reported for each task.

| | 1 Billion Tokens | | 2 Billion Tokens | | 3 Billion Tokens | |
|---|---|---|---|---|---|---|
| | Uniform | SKILL-IT | Uniform | SKILL-IT | Uniform | SKILL-IT |
| ARC Challenge (acc norm) | 35.4 | 34.6 | 35.3 | 34.9 | 34.6 | 34.8 |
| ARC Easy (acc norm) | 62.2 | 61.2 | 62.4 | 61.7 | 62.5 | 62.0 |
| BoolQ | 68.9 | 68.2 | 67.7 | 68.6 | 67.2 | 68.7 |
| COPA | 81.0 | 82.0 | 80.0 | 81.0 | 81.0 | 81.0 |
| HellaSwag (acc norm) | 63.9 | 63.7 | 63.8 | 63.9 | 64.0 | 63.9 |
| LAMBADA OpenAI | 64.4 | 67.0 | 65.9 | 66.7 | 66.8 | 66.0 |
| PIQA (acc norm) | 74.8 | 75.0 | 75.5 | 75.2 | 75.0 | 75.7 |
| Winogrande | 62.8 | 63.9 | 63.9 | 63.2 | 63.4 | 63.1 |
| Average accuracy | 64.2 | 64.4 | 64.3 | 64.4 | 64.3 | 64.4 |

### E.3.2 Weight trajectories

We provide SKILL-IT's weight trajectories for each result. The weight per skill across training steps for the LEGO pre-training experiment corresponding to Figure 4 (left) is shown in Figure 22. We see that SKILL-IT initially allocates more weight to skill 2 and less to 1,3,4,5. Since skill 1 is learned quickly, the weight on skill 1 immediately drops to below 0.1 at 1000 steps. The weight on skills 3,4, and 5 increase from around 0 to 3000 steps, during which their respective validation losses are higher than those of skills 1 and 2. Near the end of training, all losses are converging to 0, and so the weight per skill is roughly uniform.

The weight per skill across training steps for the addition pre-training experiment corresponding to Figure 4 (right) is shown in Figure 23. SKILL-IT allocates more weight to skill 2, which has an edge to skill 1 as shown in Figure 12. It also allocates very little weight to skill 3, which is learned faster than the other two skills. Eventually, it puts more weight on skill 1, the hardest skill, and then converges to uniform sampling as all validation losses approach 0.

The weight per skill across training steps for the LEGO fine-tuning experiment and the Spanish question generation and stance detection experiments corresponding to Figure 6 is shown in Figure 24. Since there is only one target skill in these experiments, the mixture of weights approaches uniform

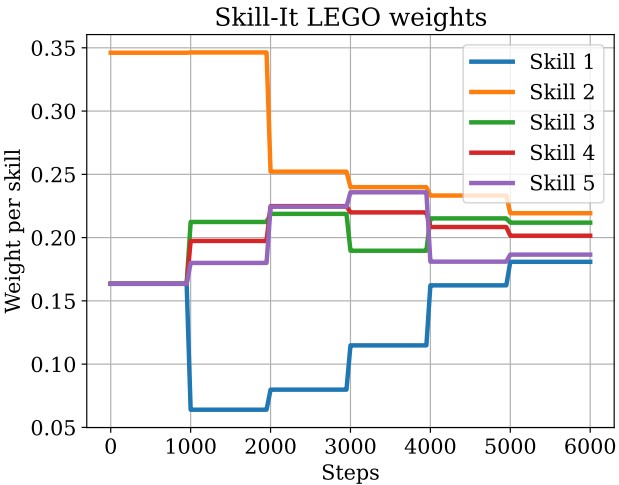

Figure 22: Weight per skill for LEGO pre-training experiment. SKILL-IT initially allocates more weight to skill 2, but eventually puts more weight on harder skills (3,4,5) before converging to uniform sampling when all losses converge roughly to 0.

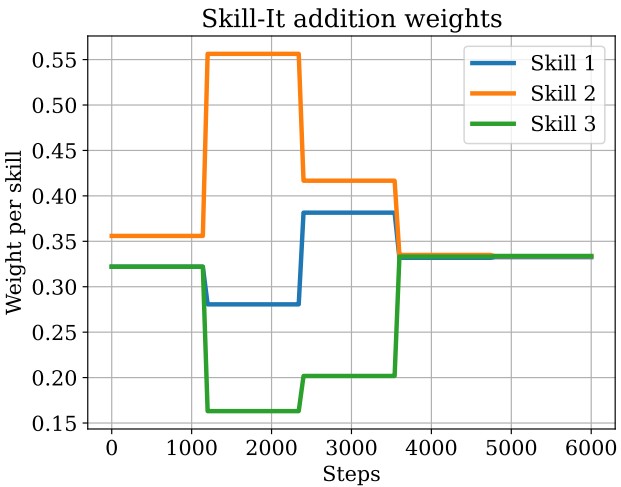

Figure 23: Weight per skill for addition pre-training experiment. SKILL-IT initially allocates more weight to skill 2, which has an edge to skill 1, while allocating little weight to skill 3 which is learned quickly. Eventually, SKILL-IT puts more weight on the harder skill 1 before converging to uniform sampling when all losses roughly approach 0.

as the loss on the target skill approaches 0. It is interesting to explore how to reduce edge weights and regularization so that the mixture approaches the target skill instead, although preliminary experiments where we decayed the edge weight and the strength of the Bregman divergence term did not appear better. We hypothesize that since training on a uniform mixture (as in Figure 3) did strictly better than training on the target skill and their loss curves did not intersect during the training run, it is better to allocate non-negligible weight on all skills throughout the training run.

The weight per skill across training steps for the Natural Instructions out-of-domain experiment corresponding to Figure 7 is shown in Figure 25, where the legend is provided for the top 10 task categories with the largest weights. While the initial weights based on the skills graph roughly establishes the order of weight magnitude, the differences among the losses on the evaluation skills increases the range of weights as training continues. As validation losses saturate, the weights also converge to fixed values.

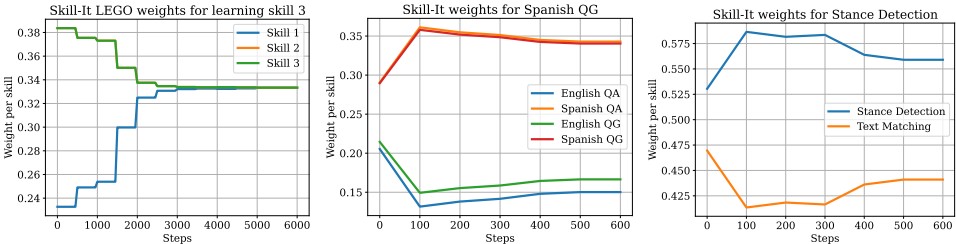

Figure 24: Weight per skill for fine-tuning experiments. Left: LEGO; Center: Spanish question generation; Right: stance detection.

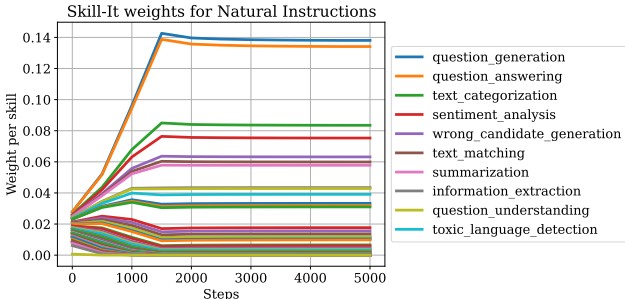

Figure 25: Weight per skill for Natural Instructions out-of-domain experiment. The legend shows the top 10 skills with the largest weight. While the relative order of weight magnitude does not change significantly across training, the incorporation of loss dramatically increases the range of the weights, showing the importance of an online algorithm.

### E.4   Experiments on 1.3B parameter model

We demonstrate that the skills graph learned on the 125M parameter model, as described in section D.2 can be used for data selection with the GPT-Neo-1.3B model across all three evaluation settings. All results are reported over 3 random seeds.

**Continual pre-training**    We report results for the LEGO continual pre-training experiment and the Natural Instructions continual pre-training experiment.

For the LEGO experiment, we use the skills graph learned on the 125M parameter model and train the 1.3B parameter model with SKILL-IT for 1500 steps with $\eta = 0.5, T = 30, w = 3$. In Figure 26, SKILL-IT still outperforms random and skill-stratified sampling on average. In particular, while performance across sampling methods is similar for early skills, the discrepancy is larger for skill 5, for which SKILL-IT allocates more weight to dynamically. In Figure 27, we provide the weight trajectories of SKILL-IT. We observe that the weight trajectories are similar to that on the 125M parameter model, where initial weight is allocated towards skill 2. Later on, more weight is allocated towards skills 4 and 5, whose losses are higher, and eventually the weight mixture converges to uniform as all losses converge to near 0.

For the Natural Instructions experiment, we use the skills graph learned on the 125M parameter model and train the 1.3B parameter model with SKILL-IT for 5000 steps with $\eta = 0.2$, $T = 1$, and learning rate $1e - 6$. In Table 10, we report performance of SKILL-IT on the Natural Instructions skills and find that the trends from the smaller model hold—SKILL-IT outperforms random and skill-stratified sampling on average.

**Fine-tuning**    We use the skills graph from the 125M parameter model to train the 1.3B parameter model with SKILL-IT for Spanish Question Generation fine-tuning. For 300 steps with learning rate $5e - 6$, $\eta = 0.8$, and $T = 12$, we find in Figure 28 that SKILL-IT outperforms both skill-stratified and random sampling.

**Out-of-domain**    We evaluate if the skills graph learned using the 125M parameter model can be used in SKILL-IT on a 1.3B parameter model for the out-of-domain setting on Natural Instructions. Our results are shown in Figure 29, where we trained for 1000 steps, learning rate $1e - 6$, $\eta = 0.5$, and $T = 2$.

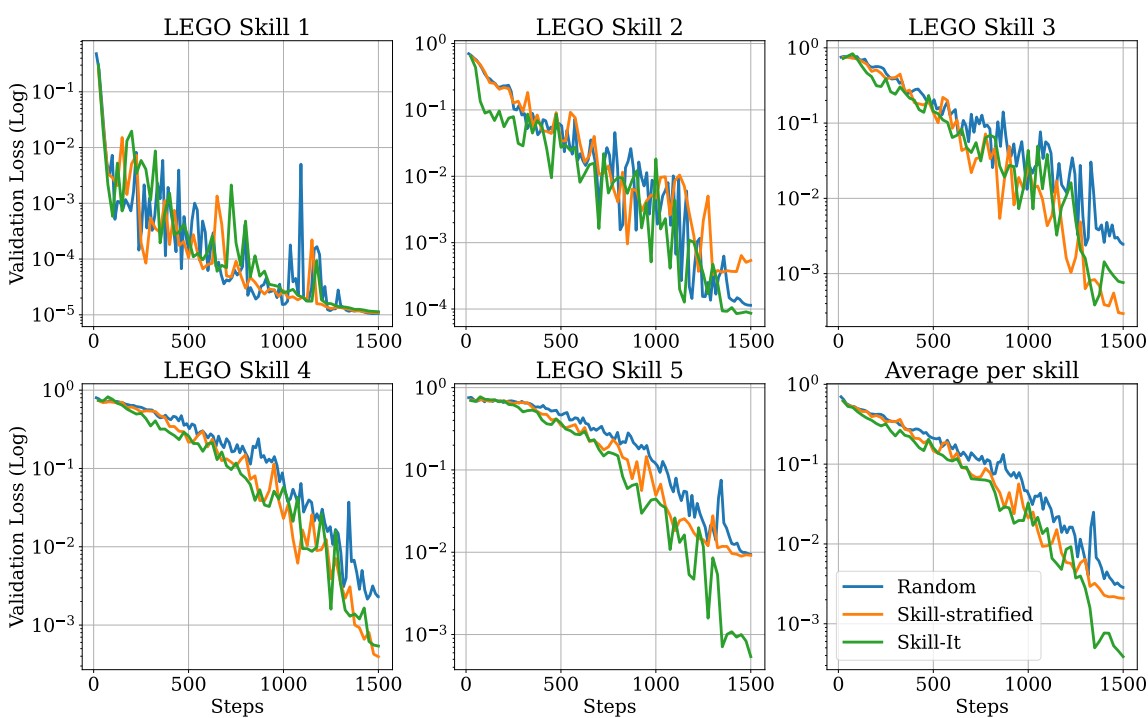

Figure 26: Performance of SKILL-IT for LEGO pre-training setting when skills graph is learned on a 125M parameter model and used for data selection with a 1.3B model. SKILL-IT on average still outperforms random and skill-stratified sampling, suggesting that findings on ordered skill sets can transfer from small models to large models.

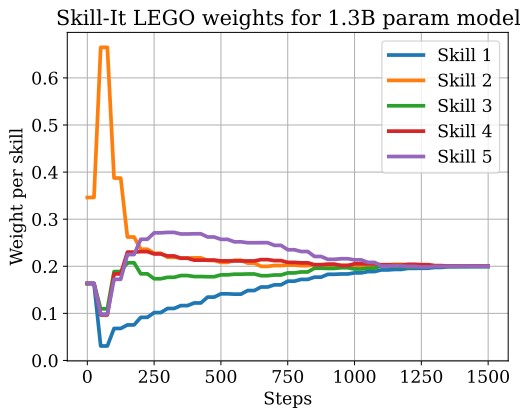

Figure 27: Weight per skill for LEGO pre-training experiment on 1.3B parameter model. The trajectories are similar to those of the 125M parameter model in Figure 22. SKILL-IT initially allocates more weight to skill 2, but eventually puts more weight on skills 4 and 5 before converging to uniform sampling when all losses converge to near 0.

Out of 12 evaluation skills, SKILL-IT outperforms random and skill-stratified sampling on 10 skills (compared to 11 skills when we used the same size model for both skills graph learning and training).

## E.5 Ablations

We report ablations on components of the skills graph learning algorithm and on other components of SKILL-IT. First, we study how the performance of SKILL-IT is robust to changes in the skills graph—learning the graph with fewer steps, using the approximate algorithm, and adding noise to

Table 10: Results when skills graph for Natural Instructions learned on 125M parameter model is used for data selection with a 1.3B model. We see that SKILL-IT on average still outperforms random and skill-stratified sampling, even though the edges used by SKILL-IT are not derived from the larger model.

| Skill | Random | Skill-stratified | SKILL-IT |
|---|---|---|---|
| Answer Verification | $2.005_{\pm 0.059}$ | $1.903_{\pm 0.069}$ | $1.890_{\pm 0.072}$ |
| Code to Text | $0.302_{\pm 0.032}$ | $0.204_{\pm 0.022}$ | $0.269_{\pm 0.032}$ |
| Discourse Connective Identification | $2.529_{\pm 0.046}$ | $2.372_{\pm 0.054}$ | $2.393_{\pm 0.056}$ |
| Entity Generation | $2.108_{\pm 0.328}$ | $1.788_{\pm 0.429}$ | $1.885_{\pm 0.461}$ |
| Entity Relation Classification | $1.130_{\pm 0.048}$ | $0.836_{\pm 0.006}$ | $0.841_{\pm 0.010}$ |
| Information Extraction | $2.032_{\pm 0.013}$ | $1.992_{\pm 0.006}$ | $1.933_{\pm 0.013}$ |
| Irony Detection | $2.802_{\pm 0.125}$ | $2.528_{\pm 0.146}$ | $2.585_{\pm 0.149}$ |
| Preposition Prediction | $1.095_{\pm 0.040}$ | $0.686_{\pm 0.041}$ | $0.774_{\pm 0.029}$ |
| Punctuation Error Detection | $2.633_{\pm 0.027}$ | $3.188_{\pm 0.055}$ | $2.726_{\pm 0.025}$ |
| Question Answering | $1.947_{\pm 0.003}$ | $2.119_{\pm 0.003}$ | $2.073_{\pm 0.001}$ |
| Question Generation | $2.214_{\pm 0.007}$ | $2.345_{\pm 0.008}$ | $2.263_{\pm 0.010}$ |
| Question Understanding | $1.928_{\pm 0.020}$ | $1.837_{\pm 0.031}$ | $1.700_{\pm 0.042}$ |
| Sentence Expansion | $2.054_{\pm 0.018}$ | $1.828_{\pm 0.060}$ | $1.853_{\pm 0.058}$ |
| Sentiment Analysis | $2.771_{\pm 0.009}$ | $2.818_{\pm 0.006}$ | $2.774_{\pm 0.007}$ |
| Stance Detection | $1.814_{\pm 0.151}$ | $1.500_{\pm 0.117}$ | $1.628_{\pm 0.149}$ |
| Summarization | $2.531_{\pm 0.009}$ | $2.472_{\pm 0.012}$ | $2.440_{\pm 0.013}$ |
| Text Categorization | $2.289_{\pm 0.016}$ | $2.341_{\pm 0.021}$ | $2.231_{\pm 0.022}$ |
| Text Matching | $1.967_{\pm 0.008}$ | $1.913_{\pm 0.005}$ | $1.872_{\pm 0.005}$ |
| Text Simplification | $1.861_{\pm 0.003}$ | $1.692_{\pm 0.023}$ | $1.698_{\pm 0.022}$ |
| Text to Code | $0.614_{\pm 0.030}$ | $0.518_{\pm 0.030}$ | $0.585_{\pm 0.022}$ |
| Toxic Language Detection | $2.853_{\pm 0.020}$ | $2.911_{\pm 0.019}$ | $2.862_{\pm 0.018}$ |
| Word Semantics | $1.999_{\pm 0.023}$ | $1.870_{\pm 0.039}$ | $1.902_{\pm 0.024}$ |
| Wrong Candidate Generation | $2.187_{\pm 0.028}$ | $2.192_{\pm 0.023}$ | $2.140_{\pm 0.020}$ |
| Average | $1.985_{\pm 0.022}$ | $1.907_{\pm 0.027}$ | $1.883_{\pm 0.032}$ |

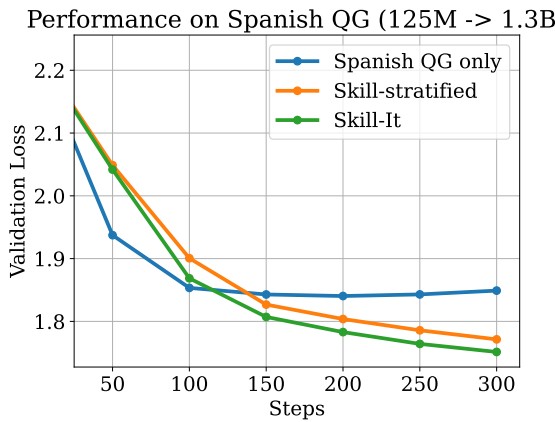

Figure 28: Performance of SKILL-IT for Spanish Question Generation fine-tuning setting when skills graph is learned on a 125M parameter model and used for data selection with a 1.3B model. SKILL-IT outperforms random and skill-stratified sampling, suggesting that findings on ordered skill sets can transfer from small models to large models.

the adjacency matrix. Second, we study the effect of removing the skills graph, using a static skills mixture, and varying the $\eta$ in SKILL-IT.

### E.5.1 Skills Graph Ablations

We explore the effect of changing how the skills graph is learned and adding noise to the graph on SKILL-IT.

First, in Appendix C.2 we propose a brute-force graph learning algorithm and an approximate graph learning algorithm. We find that these algorithms exhibit little difference in performance. For the LEGO continual pre-training experiment, we find that replacing the brute-force skills graph with an

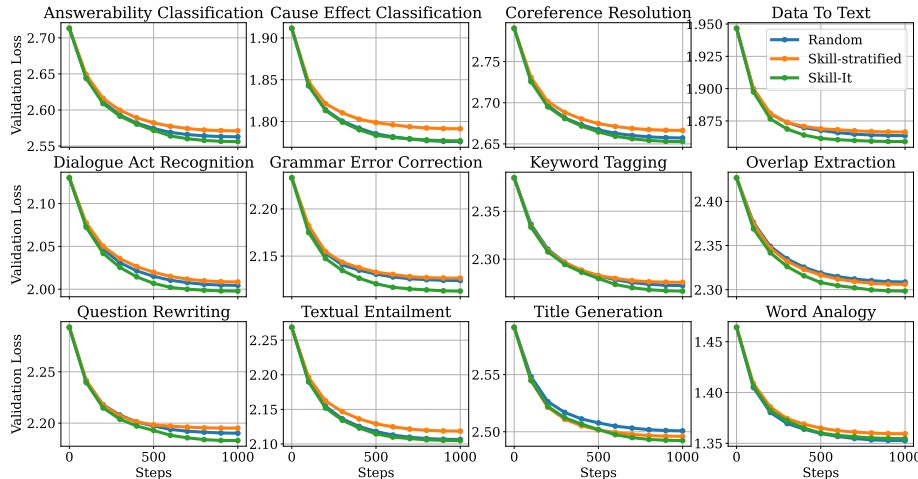

Figure 29: Performance of SKILL-IT for Natural Instructions out-of-domain setting when skills graph is learned on a 125M parameter model and used for data selection with a 1.3B model. SKILL-IT outperforms random and skill-stratified sampling on 10 out of 12 skills.

approximate skills graph yields an average validation loss of $0.017_{\pm 0.005}$ compared to $0.021_{\pm 0.009}$. For the Spanish question generation fine-tuning experiment, we find that using the brute-force algorithm yields a validation loss of $2.339_{\pm 0.028}$ compared to $2.328_{\pm 0.029}$. In both experiments, there was no more than a $0.011$ gap in validation loss from using one skills graph learning algorithm over the other, and both algorithms resulted in SKILL-IT outperforming baselines.

We next show that we can reduce $H$, the number of steps for learning each edge in the skills graph, without significantly compromising SKILL-IT's performance. In Table 11, we are able to reduce $H$ by at least $50\%$ without changing the downstream average validation loss of SKILL-IT by more than $0.01$. Results are shown for three experiments: LEGO continual pre-training, Spanish question generation fine-tuning, and Natural Instructions out-of-domain.

Table 11: Average validation loss of SKILL-IT with graph learned using original steps versus reduced steps.

| Experiment | Original steps | Reduced steps | Skill-It (original) | Skill-It (reduced) |
|---|---|---|---|---|
| LEGO PT | 6000 | 1500 | $0.021_{\pm 0.009}$ | $0.019_{\pm 0.007}$ |
| Spanish QG FT | 600 | 300 | $2.328_{\pm 0.029}$ | $2.338_{\pm 0.024}$ |
| NI OOD | 600 | 50 | $2.540_{\pm 0.003}$ | $2.538_{\pm 0.001}$ |

We also show that perturbations in the values of $A$ do not significantly impact the performance of SKILL-IT. We learn each individual skills graph using one out of 5 random seeds. We then use each skills graph as input to SKILL-IT and measure the per-element variance in the $A$ adjacency matrix as well as the variance in the average validation loss from using SKILL-IT. In Table 12, we see that the skills graph has $\mathcal{O}(1e-3)$ variance while SKILL-IT's loss has $\mathcal{O}(1e-4)$ variance. We further study the effect of perturbations by applying Gaussian noise to each element of $A$ and measuring the performance of SKILL-IT (Figure 30). We find that SKILL-IT becomes worse than baselines only after perturbing $A$ by a standard deviation of roughly $0.13$ per element, which we never attain in Table 12.

Table 12: Variance in skill graph's adjacency matrix $A$ over 5 random seeds compared with variance in SKILL-IT's performance from using different skills graphs.

| Experiment | Variance in $A$ | Variance in SKILL-IT's average validation loss |
|---|---|---|
| LEGO PT | $2.40 \times 10^{-3}$ | $4.93 \times 10^{-5}$ |
| Spanish QG FT | $1.31 \times 10^{-4}$ | $3.76 \times 10^{-4}$ |
| NI OOD | $9.22 \times 10^{-5}$ | $6.35 \times 10^{-5}$ |

Lastly, one may wonder if it is better to allocate the $\mathcal{O}(Hk)$ training steps needed for skills graph learning towards just training a model longer using a naive baseline that does not utilize skills. Let $h$ be the re-

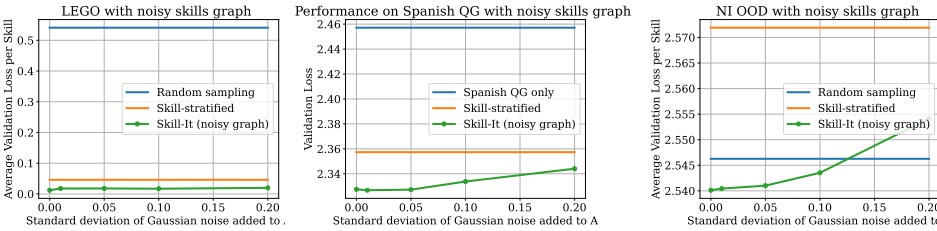

Figure 30: SKILL-IT on LEGO continual pre-training, Spanish question generation fine-tuning, and Natural Instructions out-of-domain when Gaussian noise is added to each element of adjacency matrix $A$. We use standard deviation $= \{0.01, 0.05, 0.1, 0.2\}$.

duced overall number of steps for skills graph learning from Table 11, and let $n$ be the number of steps for training the LM. We consider a random sampling baseline on $n+h$ steps and compare it to SKILL-IT on $n$ steps, with the skills graph learned over $h$ steps. In Table 13, the random baseline does not outperform SKILL-IT, suggesting that the improvement in model performance outweighs the graph learning cost.

Table 13: Comparison of SKILL-IT versus allocating skills graph learning budget to extend training with random sampling.

| Experiment | $n+h$ | $n$ | Random $(n+h)$ | Skill-It $(n)$ |
|---|---|---|---|---|
| LEGO PT | 21000 | 6000 | $0.510_{\pm 0.229}$ | $0.019_{\pm 0.007}$ |
| Spanish QG FT | 1800 | 600 | $2.430_{\pm 0.019}$ | $2.338_{\pm 0.024}$ |
| NI OOD | 8950 | 6000 | $2.541_{\pm 0.001}$ | $2.538_{\pm 0.001}$ |

### E.5.2 SKILL-IT ablations

We report ablations on removing the skills graph, using a static skills mixture, and varying the $\eta$ in SKILL-IT.

First, instead of using $A$ in Algorithm 1, we study the performance when the identity matrix is used instead; intuitively, this corresponds to a misspecified skills graph where no skill influences another skill. We refer this approach as "No graph". Note that the opposite case of a complete graph recovers skill-stratified sampling, which we already have as a baseline.

Second, instead of sampling over multiple rounds and weighting according to the loss of each skill, we study the effect of setting $T = 1$, which only uses a softmax on $A$ to yield static weights on the skills. We refer to this approach as "Static". We omit results on Natural Instructions continual pre-training, since SKILL-IT uses $T = 1$ and using no graph with $T = 1$ recovers skill-stratified sampling. Intuitively, we expect the static version of SKILL-IT to perform somewhat well unless there is significant discrepancy among the losses (e.g. in synthetics where the loss on one skill can be close to $0$ while the other is not, versus in Natural Instructions where all losses decrease consistently). For both ablations, we sweep over values of $\eta = [0.1, 0.2, 0.5, 0.8]$.

Figure 31 shows the comparison between SKILL-IT and no graph on the continual pre-training LEGO experiment, and Figure 32 shows the comparison between SKILL-IT and a static approach. We see that both the graph and the online dynamics of SKILL-IT are important for its performance. In particular, using no graph results in allocating significant weight to harder skills early on, even though many of them have easier prerequisite skills (such as skill 3 having edges to skills 1 and 2). Using a static graph results in consistent allocation of significant weight to prerequisite skills even after their validation losses converge to near $0$, and thus the harder skills that have higher loss are not learned quickly afterwards.

We perform the same ablation on the Addition dataset—the results for this are shown in Figures 33 and Figure 34. We find that these simple baselines, including using a static graph and no graph perform similarly to SKILL-IT on average across all skills—while SKILL-IT performs the best on skill 2 compared to vanilla multiplicative weights, and SKILL-IT performs the best on skill 1 compared to a static graph. This suggests that Addition is somewhat easier than the other datasets that we consider, as SKILL-IT still outperforms other baselines, as shown in Figure 4.

Figure 35 compares SKILL-IT, no graph, and static data selection for the LEGO fine-tuning experiment. No graph can be interpreted as allocating equal weight to all training skills not equal to the target skill,

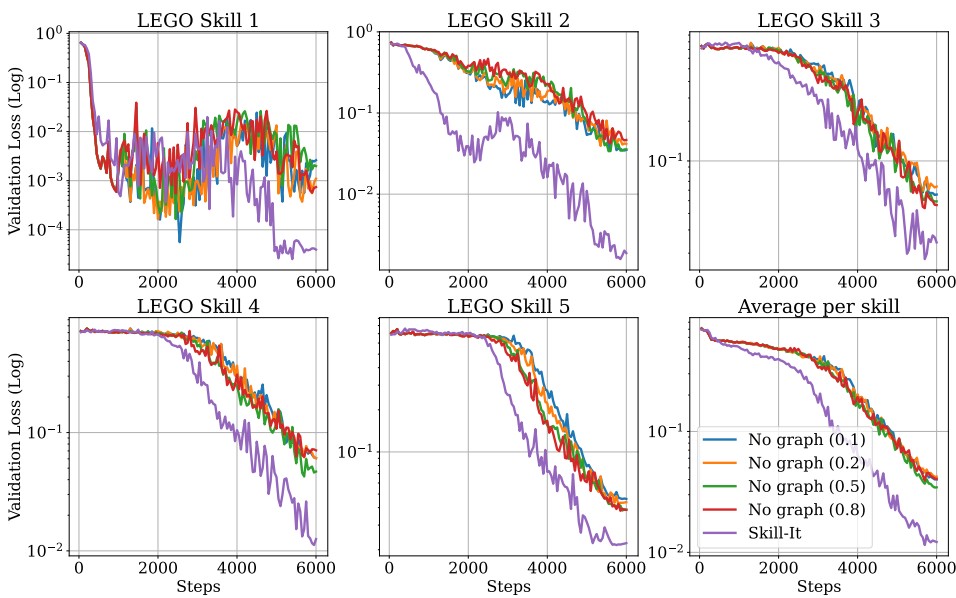

Figure 31: Comparison of SKILL-IT versus using the identity adjacency matrix (no skills graph) with $\eta = 0.1, 0.2, 0.5, 0.8$ on the LEGO continual pre-training experiment. The latter does not capture the relationship between skills, and we find that SKILL-IT attains lower loss on all skills.

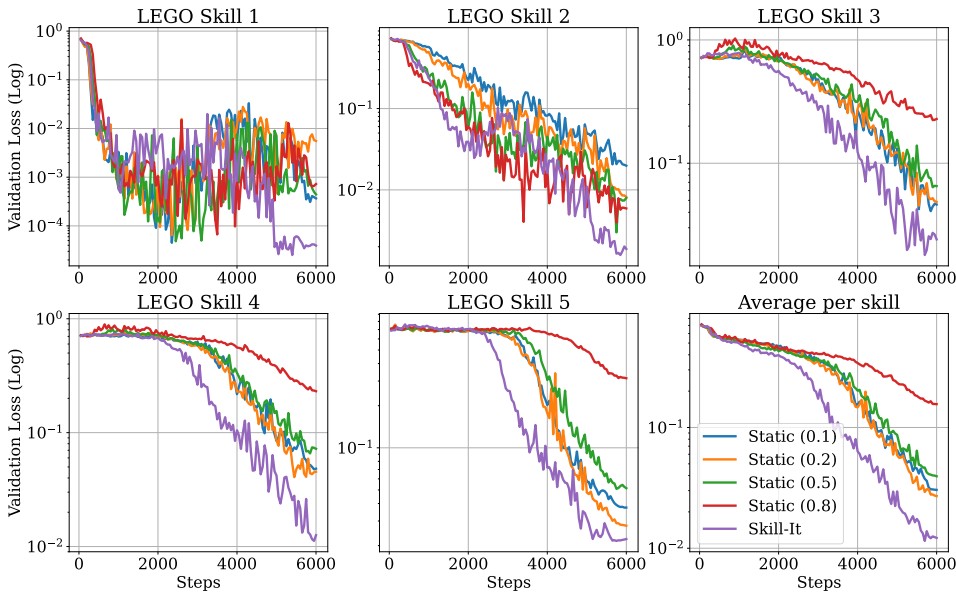

Figure 32: Comparison of SKILL-IT versus using static data selection ($T = 1$) with $\eta = 0.1, 0.2, 0.5, 0.8$ on the LEGO continual pre-training experiment. While SKILL-IT eventually allocates more weights to skills 3, 4, 5, which have higher loss, the static approach is not able to do this. We find that SKILL-IT attains lower loss on all skills.

and varying this weight versus the weight on the target skill. While SKILL-IT and setting $T = 1$ behave similarly, we see that SKILL-IT is slightly better than using no graph. For instance, SKILL-IT obtains a validation loss of 0.05 in 2000 steps, compared to 2050-2200 steps when using no graph.

Figure 36 and 37 compare SKILL-IT, no graph, and static data selection for the Natural Instructions fine-tuning experiments. For both Spanish QG and stance detection, SKILL-IT attains lower loss than using no graph or using $T = 1$ round.

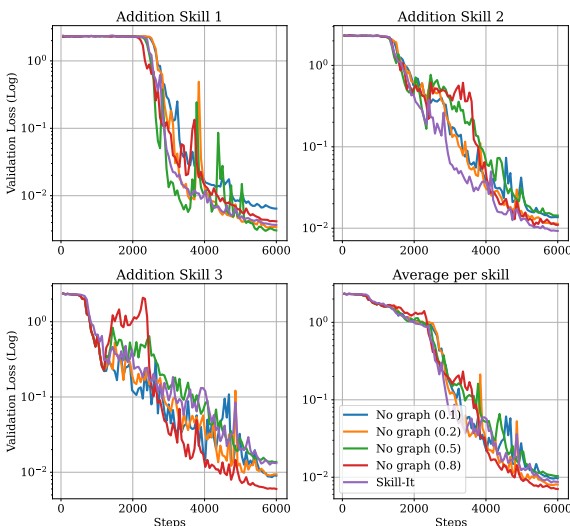

Figure 33: Comparison of SKILL-IT versus using the identity adjacency matrix (no skills graph) with $\eta = 0.1, 0.2, 0.5, 0.8$ on the Addition continual pre-training experiment. The latter does not capture the relationship between skills, and we find that SKILL-IT attains lower loss on skill 2, but attains similar performance to methods that do not use the skills graph.

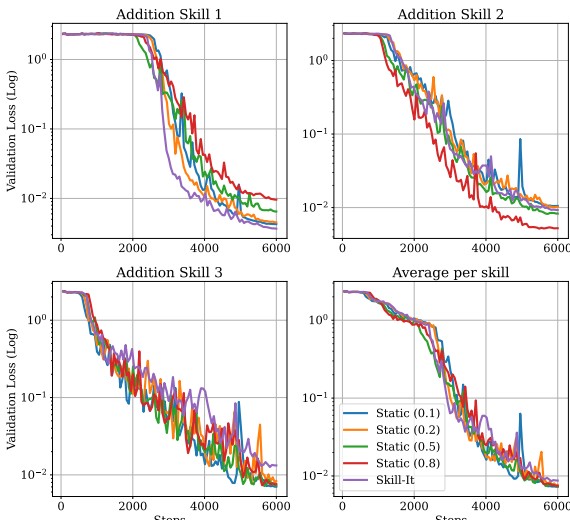

Figure 34: Comparison of SKILL-IT versus using static data selection ($T = 1$) with $\eta = 0.1, 0.2, 0.5, 0.8$ on the Addition continual pre-training experiment. We find that SKILL-IT attains lower loss on skill 1, but attains similar performance to the static methods.

Figure 38 compares SKILL-IT and static data selection for the Natural Instructions out-of-domain experiment. SKILL-IT attains the lowest validation loss on 7 out of 12 evaluation skills. It has an average loss of $2.540$ compared to a range of $2.541$-$2.551$ for static data selection.

Lastly, we study the effect of varying $\eta$ in SKILL-IT. We use $\eta \in \{0.1, 0.2, 0.5, 0.8\}$. For LEGO continual pre-training and Spanish question generation fine-tuning (Figure 39), we find that SKILL-IT with varying $\eta$ outperforms skill-stratified and random sampling (with the exception of $\eta = 0.1$ and skill-stratified being roughly the same for Spanish question generation). On the Natural Instructions out-of-domain experiment, we find that for $\eta = 0.1$ and $0.2$, we outperform baselines on $9$ and $11$ out of $12$ evaluation skills, respectively, while increasing $\eta = 0.5$ and $0.8$ resulted in outperforming on $5$ and $3$ skills. These findings show that SKILL-IT is generally not hypersensitive to the choice of $\eta$, and we hope they provide some guidelines for a reasonable range of $\eta$ without requiring significant tuning.

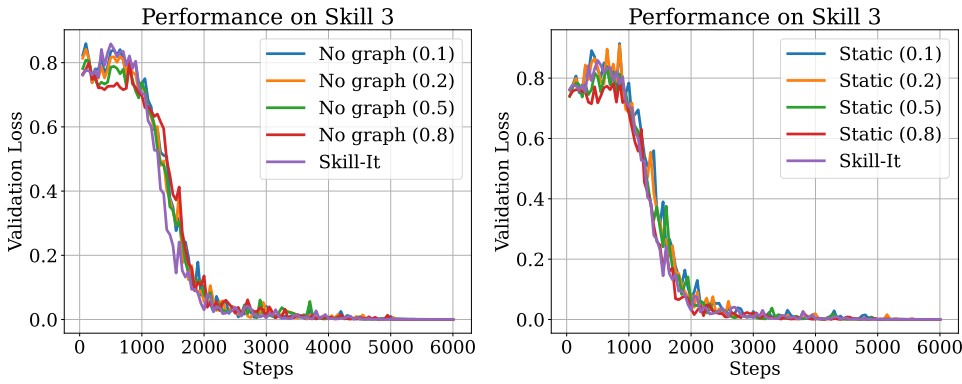

Figure 35: Comparison of SKILL-IT versus using no graph (left) and static data selection (right) with $\eta = 0.1, 0.2, 0.5, 0.8$ on the LEGO fine-tuning experiment. All approaches have roughly the same loss trajectories, but SKILL-IT is slightly lower than using no graph.

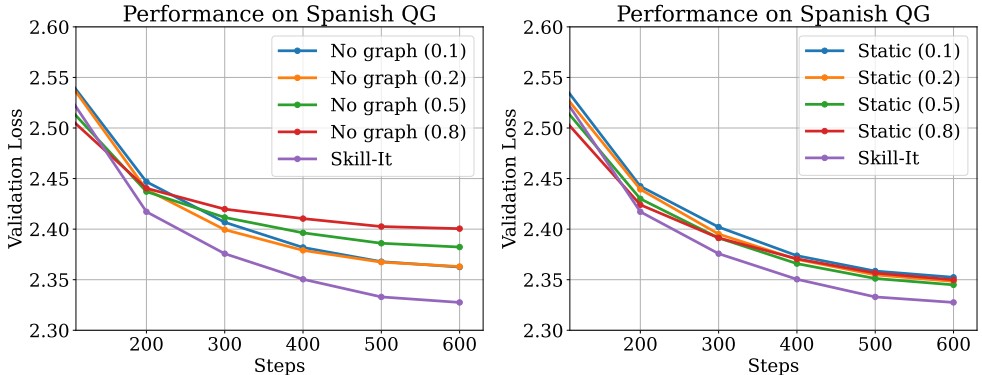

Figure 36: Comparison of SKILL-IT versus using no graph (left) and static data selection (right) with $\eta = 0.1, 0.2, 0.5, 0.8$ on the Natural Instructions Spanish QG fine-tuning experiment. SKILL-IT attains lower validation loss than both no graph and static data selection.

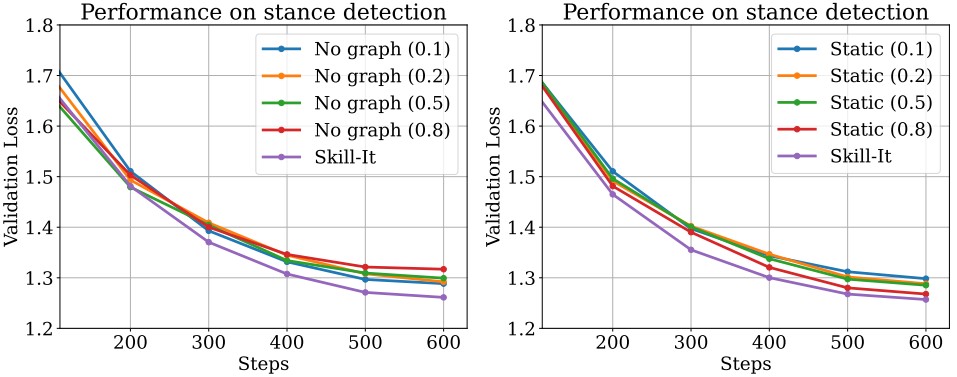

Figure 37: Comparison of SKILL-IT versus using no graph (left) and static data selection (right) with $\eta = 0.1, 0.2, 0.5, 0.8$ on the Natural Instructions stance detection fine-tuning experiment. SKILL-IT attains lower validation loss than both no graph and static data selection.

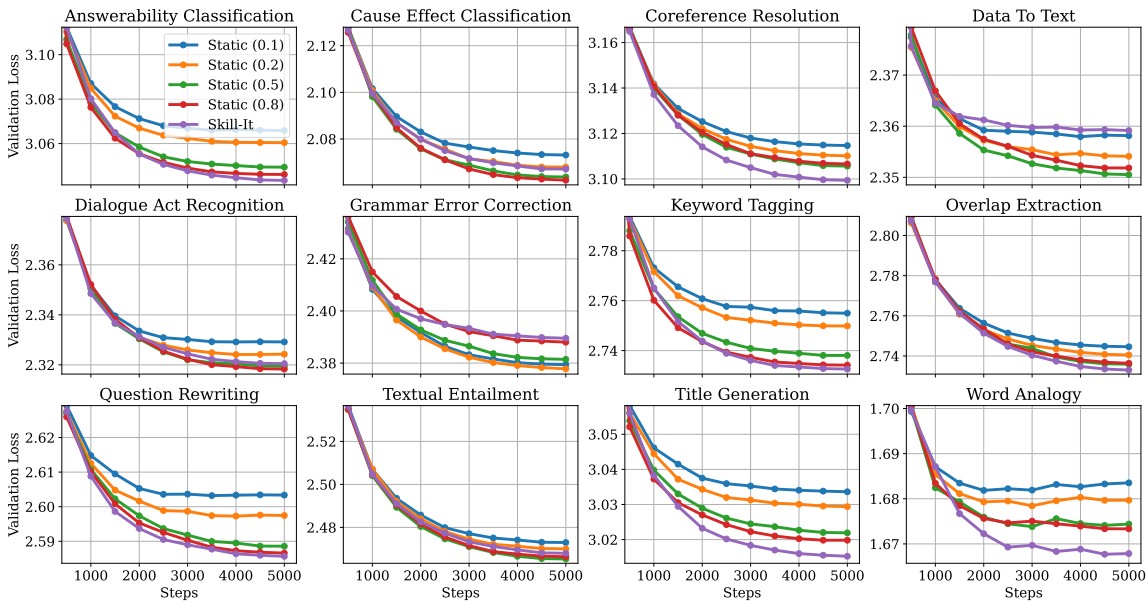

Figure 38: Comparison of SKILL-IT versus using static data selection with $\eta = 0.1, 0.2, 0.5, 0.8$ on the Natural Instructions out-of-domain experiment. SKILL-IT attains the lowest validation loss on 7 out of 12 evaluation skills, and an average loss of 2.540 compared to a range of 2.541-2.551 for static data selection.

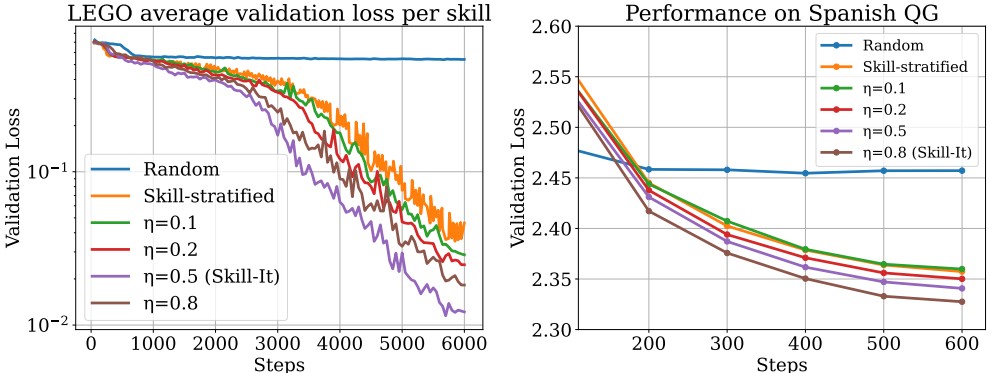

Figure 39: Skill-It ablation of $\eta \in \{0.1, 0.2, 0.5, 0.8\}$ for LEGO continual pre-training and Spanish question generation fine-tuning.

