# OpenReview forum: "Skill-it! A data-driven skills framework for understanding and training language models"
_NeurIPS.cc/2023/Conference — NeurIPS 2023 spotlight_

### Official Review · Reviewer_XvRR · 2023-06-20

**Soundness:** 3 good
**Presentation:** 3 good
**Contribution:** 3 good
**Rating:** 7
**Confidence:** 3

**Summary:**

This paper presents a way to introduce a curriculum of tokens to train LLMs. To do that, the paper introduces the concept of skills, i.e. behaviors that can be acquired by training the LLM in a subset of data, and ordered skill sets: partial orderings of skills that can be learned faster when using the skill preconditions. The paper shows that both in real and synthetic datasets, following the skill order allows for more efficient learning. Next, the paper presents techniques to sample data points from skills, given an ordered skill set. The apporach is evaluated on different datasets and model sizes, obtaining significant reductions in validation loss. Finally, the paper presents an approach to learn the skill ordering.


**Strengths:**

- Relevance and novelty: The paper addresses an important problem; given the computational cost of training LLMs, optimizing performance given a fixed compute budge is an important problem. It also presents a new framing to do so, that departs from curriculum learning approaches and focuses in selecting data to learn a set of skills. The approach to sample skills is also novel and well thought, dealing with both sampling data that both improve the weaker skills and selects the amount of data to sample from each skill.

- Method: Starting from a reasonable optimization objective and constraint (reduce the validation loss on every skill at every round), the method derives a closed from sampling rule, which is fast and simple to compute. The proposed rule can be applied to multiple cases: 1) those where all the training skills have to be evaluated, 2) cases where only a subset are considered for eval and 3) cases where no data of the eval skill is provided during training.

- Generally (see questions) clear paper writing, the definitions and discussion on how different kinds of graphs affect good sampling strategies provides great intuition for the rest of the paper. The walkthrough examples in section 3.2 also clarifies the intuition of skill curriculum.

- While the graph learning approach is computationally expensive, it can be applied to subsets of the dataset, allowing to transfer the task dependency matrix to the full data for sampling.

- Experimental results: The method is tested across different datasets and model sizes. The proposed approach improves upon different sampling baselines, including methods that sample uniformly across skills, by obtaining lower validation losses or comparable losses than the baselines in a shorter amount of time. The method also improves over baseliens on out of domain settings, where the test skills are not available during training.

**Weaknesses:**

- The method relies very strongly on having a task dependency graph. While the paper proposes two approaches to obtain these graphs, I would like further discussion into that, particularly on how to obtain the edge weights, and the robustness in the appraoch to errors or noise in this depnedncy graph.

Clarity on writing:
- Definitions: In the definition of skill it would be good to clarify what the improved metric is with respect to. Is it compared to training from a random subset of X?
- Similarly, it only becomes clear in Section 4.1 that the goal is to learn a subset of skills (which can include only 1 skill or the full skill training set). It would be good to clarify this better in the introduction.


**Questions:**

- On the definition skill, it would be good to clarify what the improved metric is with respect to. Is it compared to training from a random subset of X?
- I understand from 4.4 how to learn the presence or absence of edges in the graph, but how do we find from there the weight of that edge? How sensible is the present algorithm to the edge weight.

**Limitations:**

No limitation is mentioned in the paper.

---

> ### Author Rebuttal · Authors · 2023-08-10
>
> Thank you so much for your review! We are glad that you found the paper to be novel and address an important problem. We respond to your questions below.
>
> **Robustness to errors or noise in dependency graph:** In our global response, we show that the performance of Skill-It across different random seeds of the skills graph adjacency matrix $A$ has low variance O(1e-4), and we analyze the performance of perturbed $A$, finding that element-wise perturbations with standard deviation less than 0.13 still result in Skill-It outperforming all baselines.
>
> **Clarity:** The improved metric is with respect to the original LM’s validation loss on the skill; that is, we characterize a skill as a set of data where validation loss on a held out subset of the skill goes down over time when we train on data associated with that skill. We will clarify this in the final paper. We will also clarify in the introduction that our goal is to learn a subset of skills, not necessarily the entire training skill set.
>
> **Edge weight:** When using the full algorithm (Algorithm 2), the edge weight is obtained by taking the difference between [validation loss on skill j when training for K steps on skill j] and [validation loss on skill j when training for K steps on skill i and j uniformly]. When using the approximate graph learning algorithm (Algorithm 3), the edge weight is obtained by taking the difference between [validation loss on skill j using the original LM] and [validation loss on skill j after training the LM for K steps on skill i]. More information about setting edge weights can be found in Appendix C.2 and D.2, and we will clarify this more in our paper.

---

### Official Review · Reviewer_6a64 · 2023-06-30

**Soundness:** 4 excellent
**Presentation:** 4 excellent
**Contribution:** 4 excellent
**Rating:** 8
**Confidence:** 5

**Summary:**

This paper addresses the problem of how to order the training data for a LM.  The authors define the notion of a skill, show that skills can be (noisily) identified in training data, show that skills can be ordered, and show that sampling from the data defining a skill in an intelligent manner can improve pretraining and finetuning.  They provide both batch and online training algorithms.  They present moderately comprehensive experimental results demonstrating the effectiveness of the approach.

**Strengths:**

I really liked this paper. I think it's a great piece of work, and deserves to be published.

+ The paper addresses an important and timely problem of great interest to a large community
+ The paper is well-written and polished
+ The paper strikes an excellent balance of notation, high-level ideas, and solid empirical work
+ The resulting algorithm seems easy to implement and empirically effective
+ I enjoyed the derivation of the Skill-IT algorithm - a tidy piece of work.  Well done.

**Weaknesses:**

- It's not clear exactly how skills can be effectively identified. Right now, the paper uses various forms of meta data, but this doesn't seem like quite the best (or most general) approach.  Maybe something based on the entropy of the text?


**Questions:**

None

**Limitations:**

There is no dedicated section to limitations.  I do not foresee any negative societal impacts.

---

> ### Author Rebuttal · Authors · 2023-08-10
>
> Thank you so much for your positive review! We are happy that you enjoyed the paper and found it to be well-written.
>
> **How to identify skills:** We agree that identifying skills from data is a challenging problem and an exciting direction we are studying next. We will add this problem to a future directions section of our final paper. Currently, given a dataset and a partition of it into candidate skills, our paper can learn the skills graph, identify if the graph is meaningful (otherwise, if the partition gives you a complete or empty graph, you are better off using random or stratified sampling) and then translate the skills graph into a sampling algorithm. Our preliminary results on unsupervised skills recovery in section 3.3 also suggest that clustering embeddings doesn’t always recover skills, at least for LEGO, but clustering based on validation losses of different model checkpoints trained on different mixtures is promising.

---

### Official Review · Reviewer_paFF · 2023-07-06

**Soundness:** 3 good
**Presentation:** 3 good
**Contribution:** 3 good
**Rating:** 7
**Confidence:** 4

**Summary:**

The authors introduce a new way of curriculum learning that is thematically similar to active learning: subsets of the data are broken up into "skills" and in cases where there is a dependency structure of skills, learning precursor skills can help improve performance. When the dependency structure is either complete or empty there is not huge motivation for such approaches (as they layout), but cases where there is a clear dependency structure, they find gains in both final model performance and learning speed. They test GPT-NEO on synthetic tasks that require variable tracking and addition, as well as Natural Instructions. They demonstrates these gains in both pre-training and fine-tuning settings.

**Strengths:**

0. This paper brings together some ideas on how to better train and fine-tune models on related tasks. They introduce a simple but clean (and flexible-enough) formalism of a skill that can be applied in an NLP setting.
1. They demonstrate their method over 3 datasets.
2. If one knows the graph of skills then it seems like this method will tend to be quite helpful in practice. (If not: see my concerns below!).
3. I like Figure 1, making it a tiny bit bigger with bigger fonts would be awesome!

I like this paper and I think it could be of practical use to people.

**Weaknesses:**

0. I'd prefer if more space were dedicated to talking about the adjacent matrix. It seems learning it is extremely expensive. How effective is the approximate approach? If we include the learning costs of the adjacency matrix (i.e., give all that extra training to uniform sampling), do the other approaches equal out? I think there are scenarios where we have a clear idea regarding the dependency structure and order.
1. If the adjacency matrix is slightly wrong or noised, how does that impact the results of skill it? How consistent is the matrix over different random seeds / approximations?
2. Continuing on, if there is a large quadratic operation to find the adjacency structure, where each cell basically asks, does learning task i help task j, is there any sense of which this paper is begging the question? I think not, because you're finding a practical and specific way of when and which tasks ia, ib, ic to sample to improve j.

Nit:
1. I think the presentation of the LEGO example could possibly be made better, but I don't have great suggestions.
2. The charts are not consistent in sizing. It is tough but doing so would improve presentation.

**Questions:**

0. I think this paper could be greatly improved with some deeper analysis in how the algorithm ends up actually working. Where and how does it differ from a straightforward curriculum? Even for a simple case like addition, are we mostly learning add1 and then add2 and then add3? Is it mixed? Do the proportions shift? Showing diagrams with the proportions + in comparison to random sampling and anti-curriculum could be illuminating. (I now see you have a motivation version of this in Figure 1. I'd really like to see the actual versions.
1. Do you have a sense why Skill-stratified did worse across the board in Figure 6?
2. How can Skill-it be better for Add-1? How is it different than random or curriculum? (To confirm: Those other cases are sampling across all the various Add-#s).
3. Perhaps I missed this, but how does active-learning compare to anti-curriculum?

**Limitations:**

Seems good.

---

> ### Author Rebuttal · Authors · 2023-08-10
>
> Thank you so much for your positive review. We are glad you liked the paper and think it could be of practical use. We address your concerns below.
>
> **Learning the skills graph:** We hope your questions around the approximate approach and the random baseline with larger training budget to adjust for the cost of graph learning have been addressed by our global response. Regarding your comment that there are scenarios where we know the dependency structure, we agree this is largely true, such as in the Spanish QG case. However, there were also cases where this did not hold. For example, in LEGO, learning Skill 1 did not help learn Skill 2 faster, even though the LEGO synthetic is designed such that Skill 2 requires completion of Skill 1. In these cases, learning the skills graph is necessary until we better understand why certain dependencies exist, which is an exciting direction for future work.
>
> **Skills graph robustness to noise:** In our global response, we show that the performance of Skill-It across different random seeds of the skills graph adjacency matrix $A$ has low variance O(1e-4), and we analyze the performance of perturbed $A$, finding that element-wise perturbations with standard deviation less than 0.13 still result in Skill-It outperforming all baselines.
>
> **Learning the  adjacency matrix efficiently:** We study three different settings: pre-training, fine-tuning and out-of-domain:
>
> - For the pre-training setting, while the brute-force method requires one to learn the adjacency matrix in a pairwise manner, we show in our global response how one can reduce the learning cost to be linear in the number of skills without affecting validation loss. Additionally, we also provide two other solutions: using a smaller LM to learn the adjacency matrix and shorter training runs to reduce the complexity of learning each edge. Both these methods reduce the computational cost by an order of magnitude with little impact on performance.
>
> - For both the fine-tuning and out-of-domain setups, we only need to learn a subset of the adjacency matrix depending on the number of evaluation skills. For fine-tuning, we only need to compute the edges from each training skill to the one target skill, and thus the graph learning cost is linear in the number of training skills.
>
> Finally, even if we use the brute-force method for the pre-training setup, this is simply an intermediate step and does not obviate the need for a sampling method for training an LM. Skill-It can thus be viewed as having two steps: 1) skills graph learning and 2) graph-aware sampling.
>
> **Analysis of Skill-It’s weight mixture versus baselines:** We have provided weight trajectories of Skill-It for LEGO pre-training, addition pre-training, and Spanish QG fine-tuning in Figure 4 of the attachment (solid lines). For LEGO pre-training and addition pre-training, the dashed lines denote the weight trajectories of curriculum learning. From these figures, Skill-It samples skills very differently from curriculum and random sampling (whose proportions are listed in Appendix E.3). We will update our draft with all weight trajectories for Skill-It and baselines.
>
> **Skill-stratified sampling in Figure 6:** In Figure 6, skill-stratified sampling is uniform over the set of prerequisite training skills that are relevant to the evaluation skills, e.g. positive rows in A. However, these skills are all sampled with equal weight, even though we found that skills like Question Answering and Question Understanding are much more influential than skills like Text to Code in terms of magnitude of edge weights.
>
> **Skill-It’s performance on Add-1:** Skill-It is better for Add-1 (ones digit addition) by exploiting the edge from Add-2 to Add-1 in the skills graph. That is, it starts out by sampling a mixture of Add-2 and Add-1, which helps learn Add-1 faster than random and curriculum baselines (see Figure 4).
>
> **Active learning versus anticurriculum learning:** Active learning and anticurriculum learning are similar in that they pick the hardest or most uncertain, and thus hopefully the most informative, samples. Active learning is typically used to determine what points to label, while anticurriculum learning is used to improve model performance. Note that in our setting, active learning is not applicable since our training objective is self-supervised.

---

> > ### Comment · Reviewer_paFF · 2023-08-18
> >
> > Thanks for your thorough global (and local!) responses. I am adjusting my review from Weak Accept to Accept.
> >
> > (I had a hard time finding the proportions listed in the appendix... I look forward to seeing the finalized draft!)

---

> > > ### Author Response · Authors · 2023-08-18
> > >
> > > Thank you so much for your response and for raising your score! We will be sure to incorporate your helpful feedback and suggestions into our final draft.

---

### Official Review · Reviewer_j55A · 2023-07-24

**Soundness:** 3 good
**Presentation:** 3 good
**Contribution:** 2 fair
**Rating:** 4
**Confidence:** 4

**Summary:**

Review Summary:

The paper explores the concept of clustering data into skills to establish a curriculum, aiming to increase the efficiency of language model (LM) training. The concept of a 'skill' is introduced as a unit of behavior with a corresponding data distribution. A skill, in this context, signifies that a model's performance improves on other data that assess a similar skill when it is trained on a dataset associated with that skill.

Attempts to use embedding-based clustering for the unsupervised recovery of the curriculum did not yield desired results. However, loss-based clustering proved to be more successful, establishing that validation loss points from the same skill tend to follow similar trajectories. These trajectories were used to cluster by loss per timesteps per run.

Equipped with this understanding, the authors suggest it is possible to generate an ordered skill set and extract a curriculum from it. The existence of such an ordered skill set in real data is demonstrated in the paper.

The paper also presents 'Skill-It', an approach to dynamically create training datasets, adjusting mixture weights based on expected improvements in generalization on the target task. However, the efficacy of this method is highly contingent on the accuracy of the transfer matrix $A$ approximation.

The method employed in the paper consists of two steps:
1. Learning $A$ via brute force or linear approximation, and by approximating $A$ with a smaller model to reduce computational cost.
2. Implementing 'Skill-It' to dynamically sample training data based on the evolution of losses.

Using synthetic and real data, the authors show that such ordered skill sets exist and can enable more data-efficient training. However, the paper also acknowledges that these ordered skill sets do not align with intuitive data groupings based on metadata and embedding clustering. Lastly, a novel online data sampling algorithm, SKILL-IT, is introduced for more efficient learning of skills. The authors demonstrate the effectiveness of their approach with improved accuracy in various training scenarios.

**Strengths:**

1. Theoretical Soundness: The paper presents a robust framework based on the innovative concept of 'skills'. The methodology, which encompasses learning an approximation of the transfer matrix $A$, skill-stratified sampling, and the implementation of 'Skill-It', appears to be well-constructed and theoretically sound.

2. Clarity of Presentation: The writing in the paper is clear, making the complex concepts and methodology accessible and comprehensible. The authors effectively communicate their ideas.

3. Potential Impact: If the proposed method is effectively evaluated and shown to significantly surpass existing baselines, this work could potentially make a substantial contribution to the field of language model training. The concepts of ordered skill sets and their application for more efficient, skill-based training could lead to improved performance of language models and open up new avenues for research in this area.

**Weaknesses:**

Methodological Issues:

1. Overlooking Computational Requirements: As the primary goal of this paper revolves around reducing computational expense, it's essential to consider the additional compute requirements of the proposed methods (e.g., graph learning part) in the evaluation of the results.

2. Emphasis on a Singular Experiment: The additional computational requirements could be offset by significant improvements in larger scale models, making the 125M to 1B experiment particularly important. However, this leaves only the results from Figure 7 / Table 8 and Table 9 as practically relevant for real-data experiments.

3. Computational Cost and Hyperparameters: The methodology is not without its complexities, including a fixed computational cost to learn $A$ and the introduction of a new hyperparameter, $\nabla$. These increase the complexity of the method, making it necessary to demonstrate a strong outperformance over the random baseline to justify their inclusion.

Experimental Shortcomings:

1. Unspecified Deviations: The authors should clarify whether the deviations reported in the tables are standard errors or standard deviations. Without this clarification, the statistical significance of the results remains uncertain.

2. Missing Data in Section 5.1: The lack of standard deviations or standard errors in the main text necessitates referring to the appendix for assessing the significance of results. For the Addition task, the paper does not sufficiently address the observed equivalence between Skill-It, Curriculum, and Anti-Curriculum. In the case of Natural Instructions (real-data), it's uncertain whether Skill-It significantly outperforms random.

3. Incomplete Reporting in Sections 5.2 and 5.3: The omission of standard errors or deviations in Section 5.2, along with the absence of a random baseline, makes it challenging to evaluate the results. Similarly, for Section 5.3 (Figure 6 / Table 7 and Figure 7 / Table 8), it's unclear whether the results are statistically significant due to the absence of reported standard errors or deviations.

4. Need for Statistical Tests in E.4: As the E.4 experiment is likely the one where the initial cost of computing $A$ can be minimized, it's critical to determine if Skill-It is significantly better than skill-stratified sampling. This requires performing appropriate statistical tests with a sufficient number of trials.

5. Absence of Hyperparameter Ablation for Skill-It: While there's an ablation study for $\nabla$ in E.5, one for Skill-It is notably absent. The authors need to demonstrate that Skill-It isn't hypersensitive to its hyperparameters, or otherwise recognize the associated costs of hyperparameter tuning.

6. Limited Comparison to Stronger Baselines: Even without being experts in curriculum learning, one may wonder why the proposed method isn't benchmarked against stronger baselines from this field. This comparison would be valuable in further establishing the strengths and weaknesses of the proposed method.

**Questions:**

See Weaknesses

**Limitations:**

The limitations aren't discussed.
I suggest some in the Weaknesses section.

---

> ### Author Rebuttal · Authors · 2023-08-10
>
> Thank you for your review! We are glad you appreciated the theoretical soundness and clarity of our paper. We address your concerns around the paper’s weaknesses below.
>
> **Computational costs and 1.3B experiments:** In our global response, we demonstrate how computational costs of the skills graph can be reduced. We also provide additional results showing that the Skill-It when trained on the 1.3B model with the graph learned on the 125M model still outperforms baselines.
>
> **Skill-It hyperparameter ablation:** We are not sure what you are referring to with $\nabla$; could you please clarify? Assuming that you are referring to $\eta$ as $\nabla$, we ablate values of $\eta = 0.1, 0.2, 0.5, 0.8$ for LEGO pre-training, Spanish QG fine-tuning, and NI out-of-domain experiments. For the former two experiments, Skill-It with varied $\eta$ consistently outperforms skill-stratified and random sampling (Figure 3). For the NI out-of-domain experiment, $\eta = 0.1$ and $0.2$ outperformed baselines on 9 and 11 out of 12 evaluation skills, respectively, while increasing $\eta = 0.5, 0.8$ resulted in outperforming on 5 and 3 out of 12 skills. These findings show that Skill-It is generally not hypersensitive to the choice of $\eta$, and we hope they provide some guidelines for a reasonable range of $\eta$ without requiring significant tuning.
>
> **Reporting deviations:** All our reported deviations are standard deviations. We will clarify this in the paper and add in the standard deviations throughout the main text rather than just the Appendix. We address your comments about individual experiments below:
>
> - Addition task: Skill-It outperforms all baselines for skill 1 (ones digit addition) and is no worse than any baseline for skill 2. However, for skill 3 we see that both Skill-It and skill-stratified sampling do worse than random, curriculum, and anticurriculum. We hypothesize this is because the addition dataset consists of the skills in the ratio of 13:14:18, and so random sampling and curriculum learning will sample skill 3 more (see Figure 4 in the attachment), thereby decreasing its validation loss.
>
> - For the difference between random sampling and Skill-It on the NI out-of-domain experiment, we ran 5 more experiments for a total of 10 trials. The average validation loss per skill with random sampling is $2.548_{\pm 0.001}$ while with Skill-It is $2.541_{\pm 0.002}$.
>
> - Figure 7/Table 8 consisted of one training run due to the scale of the experiment; nonetheless, scaling up our method is a line of future work we are working towards.
>
> - In section 5.2, the baseline is just to sample randomly from the target skill itself. We omit the random sampling baseline over the entire dataset since it is not conventional for fine-tuning—the typical approach here would be to select i.i.d. train samples corresponding to the target skill.
>
> **Need for Statistical Tests in E.4:** We compare skill-stratified sampling and Skill-It on the 1.3B model over 20 trials each. We perform an independent two-sample t-test, finding that the p-value is 0.012. This value is below the conventional threshold of 0.05, suggesting strong evidence to reject the null hypothesis that there is no difference between the means of the two methods.
>
> **Limited Comparison to Stronger Curriculum Learning Baselines:** The curriculum learning baselines in our paper (Tables 3, 4, and 6 in Appendix E.3) are commonly used to benchmark curriculum learning [4]. Moreover, our baselines span a variety of curriculum learning methods by manipulating three out of four key features that characterize these methods: the pacing function for skills-curriculum [5] (how large of a pool of top-k samples to select from per epoch), the granularity of data (sample versus skills level), and the ordering (curriculum versus anticurriculum) [3, 4]. We did not change the fourth feature, the scoring function (how to rank samples), since alternatives to model loss tend to require auxiliary information or labels [4]. We also did not incorporate a self-paced curriculum [6], which is dynamic (versus using a static model to score) but requires additional parameters to learn on each sample. We will implement more variations of curriculum learning—for instance, sweeping over hyperparameters for the pacing function as described in [4]—in our final paper.
>
> [3] Soviany et. al. Curriculum Learning: A Survey.
>
> [4] Wu et. al. When do curricula work?
>
> [5] Varshney et. al. Let the Model Decide its Curriculum for Multitask Learning.
>
> [6] Jiang et. al. Self-Paced Curriculum Learning.

---

> > ### Comment · Area_Chair_9Cag · 2023-08-14
> >
> > Thanks everyone! Reviewer j55A, the author response includes new information about computational costs, hyperparameters, and statistical significance in response to concerns raised by your review. Can you please look over both the global and individual response and let us know whether these concerns have been addressed?

---

> > > ### Comment · Area_Chair_9Cag · 2023-08-21
> > >
> > > Hi reviewer j55A,
> > >
> > > As we enter the final stage of the review process, it would be very helpful to under whether the rebuttal above addresses your concerns---please let us know!

---

### Author Rebuttal · Authors · 2023-08-10

We thank the reviewers for their valuable feedback, which we have incorporated into our draft. We are glad that reviewers found the paper to address an important and timely problem (6a64, XvRR) and our method to be novel (XvRR), clean (paFF, 6a64), and sound (j55A). Reviewers thought the approach could be very useful in practice for training LMs (j55A, paFF), and found the paper to have clear presentation (j55A, 6a64, XvRR) and consist of solid empirical work (6a64).

In our global response, we address reviewers’ questions around the computational costs of learning the skills graph (j55A, paFF, XvRR). For context, in our paper we ask the question, “does there exist an ordered skill set/skills graph that can result in substantial improvements over random sampling?” Despite convention suggesting that random sampling is effective, our work uncovers the existence of such ordered skill sets and also proposes an algorithm to learn their skills graphs in a way that isn't combinatorial (e.g., trial-and-error of task/data order). Optimizing the runtime of the skills graph algorithm was a secondary objective, so we did not extensively explore this in our submitted paper. Nonetheless, we believe the existence of ordered skill sets and a tractable skills graph learning algorithm make the paper interesting grounds for future work.

**Reducing skills graph learning algorithm’s cost (j55A, paFF):** There are several avenues for reducing the cost of the skills graph learning algorithm; our updated draft includes the following proposed refinements to the brute-force algorithm: 1) reducing the number of training steps per edge, 2) learning the graph on a smaller LM, and 3) using the approximate method that is linear in m, the number of skills. These refinements are not only practical techniques for reducing costs but also suggest that the signal needed to identify edges might lie in more accessible sources of information than the brute-force method. We empirically evaluate the effects of these improvements:

- Reducing # of steps: in Table 1 in our attachment, we are able to reduce the number of steps used to learn each edge of the skills graph by at least 50% without changing the downstream average validation loss by more than 0.01. Results are shown for three experiments: LEGO PT (pre-training), Spanish QG FT (fine-tuning), NI OOD (out-of-domain).

- Using a smaller LM (j55A): in section E.4, we learn the skills graph on GPT-Neo-125M and use it to train the GPT-Neo-1.3B model for LEGO and NI PT. In our attached update, Figure 1 includes parallel experiments on Spanish QG FT and NI OOD, where Skill-It (with original hyperparameters) results in 5.3% improvement over random for Spanish QG and outperforms baselines on 10 out of 12 evaluation skills for NI.

- Approximate method (paFF): the approximate and brute-force algorithms exhibit little difference in performance—no more than 0.011 gap in validation loss—for LEGO PR and Spanish QG FT, with both approaches outperforming baselines. For LEGO PT, using the approximate algorithm yields an average validation loss of $0.017_{\pm 0.005}$ compared to $0.012_{\pm 0.008}$ (brute-force). For Spanish QG FT, using the brute-force algorithm yields a validation loss of $2.339_{\pm 0.028}$ compared to $2.328_{\pm 0.029}$ (approximate).

**Skill-It’s robustness to noise in skills graph (paFF, XvRR):** Our findings above illustrate minimal performance degradation for Skill-It from cost reduction techniques. We extend our investigation by studying Skill-It’s overall robustness to noise in the skills graph. In Table 2, we report the average element-wise variance in A, the adjacency matrix, over 5 random seeds to learn edges. We then report the variance in the overall performance of Skill-It using these A’s. The skills graph has O(1e-3) variance, while Skill-It's average validation loss has O(1e-4) variance. In addition, Figure 2 shows how model performance changes as Gaussian noise is added element-wise to $A$. Skill-It becomes worse than skill-stratified sampling only after perturbing A by a standard deviation of roughly 0.13, which we never attain in Table 2.

Table 2
| Experiment | Variance in A | Variance in Skill-It average val loss |
| ----------- | ----------- | ----------- |
| LEGO PT |0.0024| 4.93e-05 |
| Spanish QG FT | 0.000131| 0.000376 |
| NI OOD | 9.22e-05|6.35e-05 |

**LM training vs graph learning tradeoffs (j55A, paFF):** Thank you for the suggestion. We study the effect of allocating the skills graph learning budget towards extending LM training using a naive baseline that does not utilize skills, and we have updated the paper with these results. Let h be the reduced overall number of steps for graph learning from Table 1 and let n be the number of steps for LM training. We consider a random sampling baseline for n+h steps and compare it to Skill-It for n steps, with the graph learned for h steps. In Table 3, the random baseline does not outperform Skill-It in terms of average val loss, suggesting that the improvement in model performance from Skill-It outweighs the graph learning cost. Lastly, note that several existing approaches [1, 2] require training a model to determine how to select data; however, they do not consider this additional cost in their baselines.

Table 3
| Experiment | n+h | n | Random (n+h) | Skill-It (n) |
| ----------- | ----------- | ----------- | ----------- | ----------- |
| LEGO PT | 21000 | 6000 | $0.510_{\pm 0.229}$ | $0.019_{\pm 0.007}$ |
| Spanish QG FT | 1800 | 600 | $2.430_{\pm 0.019}$ | $2.338_{\pm 0.024}$ |
| NI OOD | 7950 | 6000 | $2.541_{\pm 0.001}$ | $2.538_{\pm 0.001}$ |


As suggested by reviewers, we address the limitations of this paper, including skills graph learning, in Appendix A of the current draft.

[1] Toneva et. al. An Empirical Study of Example Forgetting during Deep Neural Network Learning.

[2] Paul et. al. Deep Learning on a Data Diet: Finding Important Examples Early in Training.

---

### Decision · Program_Chairs · 2023-09-21

**Decision:**

Accept (spotlight)

**Comment:**

This paper proposes an approach to curriculum learning in LMs. This consists of two pieces. First, a method for clustering data points into skills and computing dependencies between them, both of which can be performed automatically based on training dynamics in small models. Second, a method for automatically generating skill curricula from this dependency graph. Reviewers were on the whole quite enthusiastic about the paper; concerns about baselines and variability across runs were satisfactorily addressed by the author response.